# Extreme parsimony in ATP consumption by 20S complexes in the global disassembly of single SNARE complexes

Changwon Kim [1,7], Min Ju Shon [1,2,7], Sung Hyun Kim [1,6], Gee Sung Eun [1], Je-Kyung Ryu [3,6], Changbong Hyeon [4], Reinhard Jahn [5] & Tae-Young Yoon [1✉]

Fueled by ATP hydrolysis in *N*-ethylmaleimide sensitive factor (NSF), the 20S complex disassembles rigid SNARE (soluble NSF attachment protein receptor) complexes in single unraveling step. This global disassembly distinguishes NSF from other molecular motors that make incremental and processive motions, but the molecular underpinnings of its remarkable energy efficiency remain largely unknown. Using multiple single-molecule methods, we found remarkable cooperativity in mechanical connection between NSF and the SNARE complex, which prevents dysfunctional 20S complexes that consume ATP without productive disassembly. We also constructed ATP hydrolysis cycle of the 20S complex, in which NSF largely shows randomness in ATP binding but switches to perfect ATP hydrolysis synchronization to induce global SNARE disassembly, minimizing ATP hydrolysis by non-20S complex-forming NSF molecules. These two mechanisms work in concert to concentrate ATP consumption into functional 20S complexes, suggesting evolutionary adaptations by the 20S complex to the energetically expensive mechanical task of SNARE complex disassembly.

[1] School of Biological Sciences and Institute for Molecular Biology and Genetics, Seoul National University, Seoul, South Korea. [2] Department of Physics, Pohang University of Science and Technology, Pohang, Gyeongbuk, South Korea. [3] Department of Physics, KAIST, Daejeon, South Korea. [4] Korea Institute for Advanced Study, Seoul, South Korea. [5] Department of Neurobiology, Max-Planck-Institute for Biophysical Chemistry, Göttingen, Germany. [6] Present address: Department of Bionanoscience, Kavli Institute of Technology, Delft University of Technology, Delft, the Netherlands. [7] These authors contributed equally: Changwon Kim, Min Ju Shon. ✉email: tyyoon@snu.ac.kr

Membrane fusion is an energetically demanding process in eukaryotic cells[1,2], an evolutionary constraint that has forced membrane fusion proteins to deal with large scales of free energies[3–5]. Most notably, soluble N-ethylmaleimide sensitive factor attachment protein receptors (SNAREs) form a rigid four-helix bundle, called the SNARE complex, releasing huge free energy amounting to 65 $k_B$T[6–9]. This SNARE complex assembly brings two membranes into molecular proximity and eventually overcomes the membrane fusion barriers[10,11]. The mechanical stability of the resulting rigid SNARE complex, however, poses cells with challenging tasks to disassemble these complexes to recycle individual SNAREs[7–9]. N-ethylmaleimide sensitive factor (NSF) is responsible for this daunting mechanical task[12–15], playing an essential role in sustaining membrane trafficking in all eukaryotes[14,16–18].

NSF is a ring-shaped Type 2 AAA + (ATPases associated with various cellular activities) homohexamer that consumes adenosine triphosphate (ATP) to induce the disassembly of SNARE complexes[19–22]. Thus, NSF is a motor protein that converts the chemical energy released by ATP hydrolysis into a specific type of mechanical work. Cryo-electron microscopy (EM) studies have revealed unprecedented structural details underlying NSF-mediated disassembly of SNARE complexes[23,24]. Two to four soluble NSF attachment proteins (SNAPs) bind to a single SNARE complex, serving as specialized adaptors between the SNARE complex and NSF[13,23,24]. EM structures further revealed that ATP hydrolysis in the first AAA + hexameric layer of NSF (referred to as D1) induces large conformational changes in D1 itself and in the N-terminal domains[13,23], generating combined mechanical actions of unzipping and unwinding that lead to SNARE complex disassembly[21,25]. Thus, the 20S complex, consisting of NSF, SNAPs, and the SNARE complex, is a platform on which chemical energy is converted into concentrated mechanical work.

We recently observed that NSF induces SNARE complex disassembly in a single step that exploits mainly a single round of ATP turnover[26], which in turn induces a rapid dismantling of the 20S complex and discharge of individual SNARE proteins[26,27]. This explosive, single-step disassembly of SNARE complexes by NSF seems to have evolved to prevent reassembly of any partially unzipped SNARE complexes. These observations position NSF as one of the first examples of an unfolding machine that induces a global disassembly of large protein complexes[28,29]. It also distinguishes NSF from other molecular motors and unfoldases that typically induce linear or rotational translocation in incremental steps through many cycles of ATP binding and hydrolysis[30–32].

Considering the extreme rigidity of the SNARE complexes, the global disassembly induced by NSF raises fundamental questions, most of which remain unanswered yet: how does NSF achieve its remarkable energy efficiency? Do NSF hexamers require close coordination of ATP binding and hydrolysis? How does the 20S complex disintegrate so rapidly after the global disassembly event? Answering these questions may reveal how global disassembly machines differ from processive linear and rotational translocases. Here, we have applied a wide array of single-molecule methods to clarify our understanding of the molecular architecture and ATP hydrolysis cycle of the 20S complex. Our results collectively suggest the 20S complex has evolved sophisticated mechanisms for preventing unnecessary ATP expenditure while still accomplishing the energy-expensive mechanical task of SNARE complex disassembly.

## Results
### 20S complexes induce simultaneous unraveling of four SNARE motifs on ms timescales. To track the disassembly of 20S

complexes with a higher conformational resolution, we used high-speed single-molecule magnetic tweezers that sample protein conformational changes at rates up to 1.2 kHz[33,34]. We prepared neuronal SNARE complexes consisting of syntaxin-1A (Syx), VAMP2 (or synaptobrevin-2), and SNAP-25A. We attached short DNA handles (0.5 kbp each) to the C-terminal ends of the linker regions of Syx and VAMP2 (both of which lack transmembrane domains) (Fig. 1a). We then attached these two DNA handles to the surface of magnetic beads and the surface of the polymer-coated glass, respectively. We were able to modulate the force experienced by the magnetic bead by varying the distance between the sample and a pair of permanent magnets (Fig. 1a). When we median-filtered these data to 60 Hz, we found the Allan deviation for the vertical position of tethered beads reduced to ~2 nm, a precision sufficient for following the disassembly of single SNARE complexes (Supplementary Fig. 1a, b).

We first profiled the force-dependent conformations of individual SNARE complexes in the absence of α-isoform SNAPs (αSNAPs) and NSF (Fig. 1b). Up to forces of 12 pN, the SNARE complexes including their linker regions remained "fully zipped" (the FZ state), suggesting a substantial energy barrier to initiating disassembly. When the linker regions begin to dissociate above 12 pN, the SNARE complexes gradually enter the "linker-unzipped" (LU) state. Then, at forces above 14 pN, there is another discrete state matching the prediction for a "half-unzipped" (HU) state (Fig. 1b and Supplementary Fig. 1b).

At forces above 16 pN, all the individual SNARE complexes enter two states characterized by very high levels of extension. The lower of these two states should correspond to the "totally unzipped" (TU) state, in which VAMP2 is completely unraveled from the Q-SNAREs (Syx and SNAP-25) to become unstructured (Fig. 1c and Supplementary Fig. 1b). The extension level of TU was well explained by the Q-SNARE model with unstructured C-terminal motifs (up to +2 layer) in agreement with the results from NMR experiments[6,8,35]. In the most extended state, referred to as the "unstructured coiled" (UC) state, both VAMP2 and Syx seem to be completely unraveled and SNAP-25 is dissociated from the complex (Fig. 1c). Indeed, while we found the transitions from FZ to TU to be reversible, the final TU-to-UC transition was essentially irreversible probably due to the loss of SNAP-25 (Fig. 1c). We had to lower the force below 5 pN in the presence of free SNAP-25s to reload SNAP-25 and induce SNARE complex refolding (Supplementary Fig. 1c, d).

Under the pulling scheme we used, the unfolding transition to TU—spanning a 28 nm extension increase—occurred relatively quickly with a time constant of 1.6 s at 16 pN (Fig. 1d, upper). In contrast, the final TU-to-UC transition, despite its 4 nm extension increase, had a much longer latency at the same force (Fig. 1c), yielding a total time constant of 16.5 s for the full transition from FZ to UC (Fig. 1d, lower). We attribute this long latency to inefficient dissociation of SNAP-25. In our pulling geometry where force was not applied directly to SNAP-25, dissociation would only occur when all the molecular contacts between Syx and SNAP-25 were dismantled simultaneously in a manner similar to rupture under shearing[8].

We next examined the enzymatic disassembly of single SNARE complexes by the 20S complex. To this end, we first added αSNAPs to the SNARE complex and asked whether their interactions primed the substrate for disassembly. We could discern the binding of αSNAPs (1–5 μM) from the concentration-dependent changes in the extension distributions of single SNARE complexes (Supplementary Fig. 1e). In particular, we found that αSNAP binding suppressed the half-unzipped HU state (Supplementary Fig. 1f)[36]. Our interpretation is that αSNAPs adhere tightly to the C-terminal half of the SNARE motifs, consistent with cryo-EM structures that indicated strong

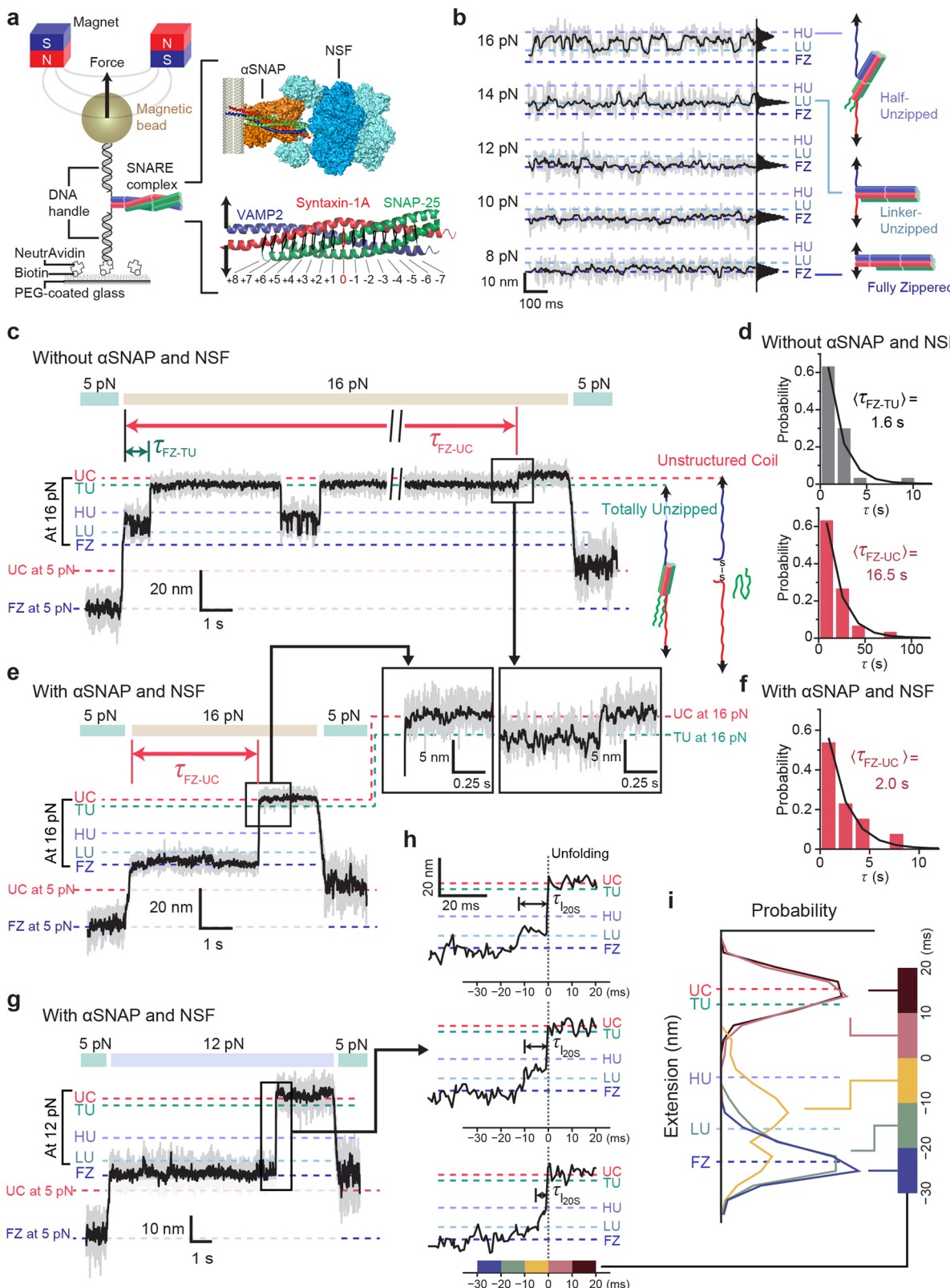

electrostatic interactions between αSNAPs and the C-terminal half of the SNARE motif[23,25].

Next, we added wild-type (WT) NSF hexamers with ATP and Mg$^{2+}$ and waited at lower forces of 2 pN for the induction of 20S complex assembly. To make a direct comparison with our mechanical unzipping data, we increased the force to 16 pN and measured the latency to unfolding events (Fig. 1e). This increase in mechanical tension accelerated SNARE complex disassembly, leading to a rapid increase in extension with a latency of only 2.0 s (Fig. 1f). Remarkably, we found that the extension traces reached the totally unstructured UC state directly without pausing at the TU state (Fig. 1e, inset), an observation in stark contrast with our force-induced unfolding results in which the final TU-to-UC transition constituted the rate-limiting transition step. Thus, in

**Fig. 1 NSF and αSNAP actively disassemble SNAREs via millisecond-long intermediates. a** Magnetic tweezers setup for observing SNARE complexes. A cartoon of membrane-bound 20S complex is shown for comparing the force-acting points. **b** Extension traces recorded at varying force levels with corresponding histograms. **c**, **e** Representative results from force-jump experiments without (**c**) and with (**e**) αSNAP and NSF (1 μM each). Both experiments were carried out with ATP (2 mM) and MgCl$_2$ (10 mM). **d**, **f** Distributions of latency to unzipping (FZ to TU) and unfolding (FZ to UC) in force-jump experiments without (**d**) and with (**f**) αSNAP and NSF. **g** Force-jump experiment for 20S complex-mediated disassembly at 12 pN. The arrow indicates the region magnified in (**h**). **h** Representative high-speed traces of 20S complex-mediated disassembly. $\tau_{\text{I20S}}$: intermediate state lifetime. Color code (bottom) represents the interval for the histograms shown in (**i**). **i** Extension distributions before and after the moment of disassembly collected from 26 disassembly events. The extension values were aligned with UC at zero for simplicity. Throughout the figure, the gray and black traces are 1.2-kHz raw and 60-Hz-filtered traces, respectively. FZ fully zippered, LU linker-unzipped, HU half-unzipped, TU totally unzipped, UC unstructured coil.

20S complex-mediated disassembly, all four SNARE motifs, including those of SNAP-25, are efficiently disassembled in a highly concurrent manner.

Finally, using high-speed tracking of our magnetic tweezers, we attempted to capture possible intermediate states during the NSF-driven disassembly of single SNARE complexes (Fig. 1g, h). The high-speed magnetic tweezers allowed us to resolve the evolution of the extension distribution for every 10 ms window (with $t = 0$ set to the completion of disassembly) (Fig. 1h, i). For the 10-ms period right before disassembly (i.e., between $t = -10$ and 0 ms), we found an unambiguous intermediate state (referred to as $I_{20S}$), of which dwell time was exponentially distributed with a time constant of 7 ms (Supplementary Fig. 2a).

Strikingly, $I_{20S}$ corresponded neither to the HU nor the linker-unzipped LU states but was instead located midway between HU and LU (Fig. 1i). Our calculations further show that the HU state cannot explain the extension of $I_{20S}$ even with an assumption of the Q-SNARE motifs maintaining intact helical structures (Supplementary Fig. 2b, c). Based on our observation that all four SNARE motifs unravel simultaneously, we thus propose an alternative model where the four SNARE motifs are unfolded to the same layer (Supplementary Fig. 2b). According to this symmetric unfolding model, the SNARE complex would be disassembled up to the +4 layer in the $I_{20S}$ state (Supplementary Fig. 2c). We note this coincides with the region identified in the cryo-EM structures as the main binding site for αSNAPs in the C-terminal half of the SNARE complex[24,25,37]. Thus, $I_{20S}$ may represent a state in which the substrate SNARE complex is unraveled by αSNAPs with the +4/+5 layer working as a point of force application (see "Discussion").

We noticed that any short-lived intermediates below hundreds of microseconds would not be resolved in our measurement with a time resolution of 0.83 ms. We cannot exclude a possibility that the $I_{20s}$ state might reflect an averaged-out level as a result of rapid transitions between LU and HU. Nevertheless, the symmetric unfolding model remains valid, in which a resulting HU state is estimated to have Syx unfolded up to the +3 layer. Thus, the disparity between $I_{20s}$ and HU turns out to be small in the symmetric unfolding, measuring only a one-layer difference.

**αSNAP dimers are the minimal units that connect NSF to SNARE complexes**. The results of our magnetic tweezer experiments indicate that the 20S complex efficiently disassembles all four α-helices of the SNARE complex simultaneously in less than 10 ms. Our observations also point to the importance of strong connections between αSNAPs and the SNARE complex for an efficient disassembly of the mechanically rigid neuronal SNARE complexes. We hypothesized that such a robust mechanical connection should be maintained throughout the 20S complex structure. Without reliable mechanical connections, the large free energy released from NSF hexamers would be uselessly dissipated rather than productively coupled to disassembly.

We, therefore, examined the mechanical coupling at the interface between αSNAPs and NSF, which potentially involves

a large number of protomers[21,23,38]. Examining the connection on the αSNAP side first, we investigated the number of αSNAPs in each of our functional 20S complexes because cryo-EM structures report different copies of αSNAPs—two to four—in 20S complex structures[13,23,24,38]. We fused αSNAPs to the 20 kDa SNAP tags at their N-terminal (i.e., membrane-proximal) ends and labeled the fused proteins with benzylguanine (BG)-Alexa647 dyes (Supplementary Fig. 3a)[39]. The resulting SNAP-tag-fused αSNAPs showed efficient binding to vesicle-reconstituted SNARE complexes and then to NSF hexamers, thereby fully supporting the formation of 20S complexes (Supplementary Fig. 3b, c). The resulting 20S complexes also efficiently catalyzed the disassembly of neuronal SNARE complexes (Supplementary Fig. 3d, e).

By measuring photobleaching steps with single-molecule resolution, we investigated the involvement of these engineered αSNAPs in 20S complex-mediated SNARE complex disassembly (Fig. 2a, b). First, in an independent experiment using a tandem array of SNAP tags (Supplementary Fig. 4a), we determined the labeling efficiency of SNAP tags to be ~90% (Supplementary Fig. 4b, c) and used this value to obtain the deconvoluted distribution of photobleaching steps (Supplementary Fig. 4d). Next, to reduce complications from non-specific surface adsorption of αSNAPs, we introduced SNAP tags to the N-terminal ends of individual NSF subunits and labeled the NSF hexamers with BG-DY549 dyes (Fig. 2a). With this double-labeling scheme, we confined our photobleaching analysis to the co-localized spots that showed both Alexa647 and Dy549 fluorescence signals (Fig. 2a–c).

Remarkably, our analysis revealed that few populations were showing a single photobleaching step of αSNAPs that successfully formed the 20S complexes (Fig. 2d). The major populations showed two or three photobleaching steps, with 14% of them showing four photobleaching steps. Also, we found that no 20S complexes populations showed more than four photobleaching steps. These results clearly suggest that αSNAP dimers define the minimal unit in connecting the SNARE complex substrate and the NSF hexamer. Although the resolved copy numbers of αSNAPs in the 20S complexes, two to four, are consistent with cryo-EM studies, our results suggest a considerable heterogeneity in the αSNAP composition, an observation uniquely resolved with the single-molecule resolution of our assays.

**Observation of cooperativity between NSF subunits in 20S complex formation and in its disassembly function**. We next examined the subunit connectivity of NSF hexamers at the αSNAP–NSF interface. To this end, we extended our single-molecule fluorescence method to permit the dissection of subunit composition in individual NSF hexamers. We generated a mutant NSF subunit, referred to as N-MT, in which we fused the SNAP tag to the C-terminal end and inserted a PreScission cleavage site right after the N-terminal (N) domain (a.a. 1–206) that is primarily responsible for NSF's interaction with αSNAPs (Fig. 3a)[21,40]. By depriving the ATP nucleotides bound to the NSF hexamer, we induced their monomerization into individual subunits[23] and labeled each SNAP tag with BG-tagged fluorescent

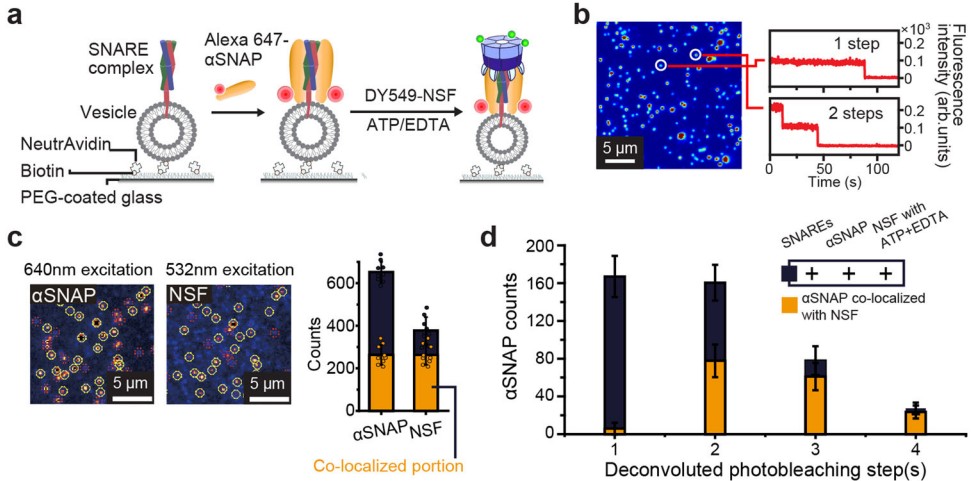

**Fig. 2 αSNAP dimers are the minimal units that connect NSF and SNARE complexes. a** Schematic for stoichiometry determination of αSNAPs bound to surface-immobilized SNARE-incorporating vesicles. **b** Representative fluorescence intensity traces for the single spots in the image with the corresponding number of photobleaching step(s). **c** Representative fluorescence images of Alexa647-labeled αSNAP and DY549-labeled NSF from a total internal reflection fluorescence (TIRF) microscope and the corresponding counts of proteins and co-localized portions. **d** Deconvoluted histogram of the photobleaching step(s) for labeled αSNAP in single spots examining the number of αSNAP for NSF binding. The histogram of photobleaching step(s) in **d** was deconvoluted accounting for the 90% labeling efficiency. Error bars in (**c**, **d**) represent mean ± s.d. for $n = 9$ images from two independent experiments (**c**) and $n = 12$ images from three independent experiments (**d**). The reproducibility of the images (**b**, **c**) was confirmed in three independent experiments. Source data are provided as a Source Data file.

dyes. After mixing monomeric NSF N-MT subunits with monomeric WT subunits at a predefined ratio, we induced re-hexamerization by adding ATP and then deleted the N-domains of N-MT subunits with PreScission proteases. Finally, we collected hexameric NSF using size-exclusion chromatography (Fig. 3a and Supplementary Fig. 5a).

To profile the N-MT subunit composition of the resulting hexamers, we induced surface immunoprecipitation (IP) of the hybrid hexamers with antibodies targeting either the N-domain or the SNAP tag (Fig. 3b). Despite using equimolar concentrations of wild-type (WT) and N-MT protomers, the deconvoluted distribution of photobleaching steps revealed a broad distribution in N-MT copy numbers (Fig. 3b), indicating the stochastic nature of re-hexamerization (Fig. 3a). We note that both IP antibodies gave almost identical results, ruling out a significant bias in our IP data specific to any individual antibody epitope (Fig. 3b).

Next, we repeated the same measurements for the 20S complexes. After loading αSNAPs onto SNARE complex-reconstituted vesicles, we completed the 20S complex assembly by adding the same pool of NSF hybrid hexamers used in Fig. 3b (Fig. 3c). When examining the photobleaching steps, we noted a marked skewing of the N-MT distribution toward hexamers with only one or two N-MT subunits (Fig. 3c). For a more quantitative description, we calculated the "relative avidity" as the ratio between the fractions observed in 20S complex populations and the fractions observed in our random sampling via surface IP (Fig. 3d). This metric drops markedly when the hexamers carry three N-MT subunits, and it practically approaches zero for hexamers with more than four N-MT subunits (Fig. 3d). These data directly suggest cooperativity between NSF subunits, in which at least three N-domains cooperate to promote stable 20S complex assembly.

We next examined the ability of the hybrid hexamers to direct disassembly. Surprisingly, the 20S complexes formed with the hybrid N-MT hexamers, 20% of which contained three N-MT subunits (Fig. 3d), were able to successfully disassemble SNARE complexes (Fig. 3e and Supplementary Fig. 5b). We increased the fraction of 20S complexes carrying three N-MT subunits to 27 % by including more N-MTs in the re-hexamerization step. Still, we

found that the disassembly activity essential maintained (Supplementary Fig. 5c, d). In addition, when examining detailed kinetics, the 20S complexes assembled with the N-MT hybrid hexamers showed identical kinetics in the disassembly reaction to those formed with the WT NSF hexamers (Fig. 3f). Together, our observations suggest two-faceted behaviors of the N-MT hybrid hexamers. Only a fraction of N-MT hybrid hexamers can form 20S complexes, especially when the hybrid hexamers carry more copies of N-MT. This small fraction that does successfully form a 20S complex, however, can disassemble SNARE complexes with a potency almost equal to that of WT hexamers.

**Identification of the core subunit connectivity within the 20S complex.** Through a precise determination of the subunit composition of individual NSF hexamers, we found that hexamers incorporating more than three N-MT subunits show a disproportionately reduced capacity for 20S complex formation (Fig. 3d). We reasoned that this suggests a geometric rule that permits the binding of certain arrangements of N-domains for 20S complex assembly while excluding others.

To determine whether such a rule exists, we examined the geometric arrangements of the N-MT subunits (Fig. 4a). There are four possible arrangements for a hexamer with three N-MT subunits (Fig. 4a, 3 N-MT cases), and the rotation of each subgroup represents all possible conformations. Except for the "ACE" subgroup, each of the remaining three arrangements represents 30% of the total arrangements. We note this value is comparable to the relative avidity (~25%) exhibited by the hybrid hexamers with three N-MT subunits. Thus, at the simplest level, we hypothesized that one of the three subgroups largely retained the capacity to form 20S complexes, while the other subgroups had lost it.

To test our hypothesis, we used the single-molecule FRET technique to dissect the relative positions of the N-MT subunits. We increased the molar ratio of the WT subunit to 60 mol% and labeled the resulting hexamers with BG-Dylight549 and BG-Alexa647 dyes (Fig. 4a, right). After inducing assembly of 20S complexes as in Fig. 3c, we included only the time-resolved

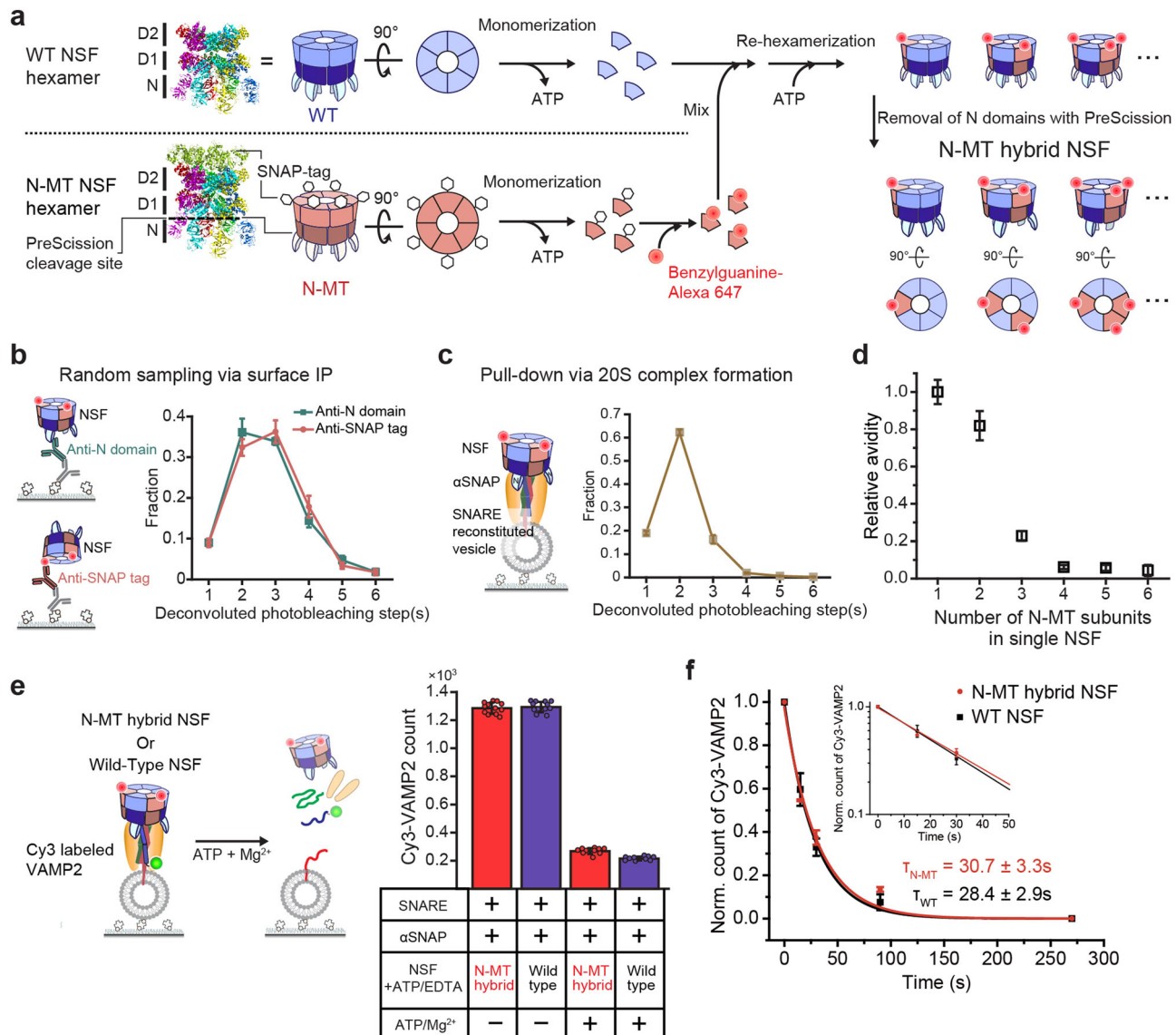

**Fig. 3 Observation of cooperativity between NSF subunits in 20S complex formation and its disassembly function. a** Preparation of N-MT hybrid NSF including its monomerization, labeling, re-hexamerization, and N-domain removal (PDB ID: 3J94[23] for NSF, 3KZZ for SNAP tag). **b** Experimental scheme of surface immunoprecipitation (IP) for the pulldown of N-MT hybrid NSF using two types of antibodies (N-domain and SNAP tag) and the distributions of deconvoluted photobleaching step(s) considering 90% labeling efficiency. **c** Experimental scheme of NSF pulldown via the SNARE–αSNAP complex and the distributions of deconvoluted photobleaching step(s). **d** Relative avidity of N-MT hybrid hexamers to the SNARE–αSNAP complex according to the number of N-MT subunits in single NSF hexamers. Data in (**c**) divided by those using the anti-N-domain antibody in (**b**) and normalized to the value for single-mutant hexamers. **e** Measurements of WT and N-MT hybrid NSF's disassembly activity. Counts for SNARE complexes under ATP-non-hydrolyzing (1 mM ATP/1 mM EDTA) and inducing ATP hydrolysis (1 mM ATP/10 mM Mg$^{2+}$) for 5 min. **f** Normalized counts of Cy3-VAMP2 under ATP-hydrolyzing condition with two types of NSF at several time points. Inset shows a semi-log plot of early time points. $\tau$ represent the time constant of exponential decay. The data represent mean ± s.e.m. for four independent experiments (**b**, **c**). The data in (**d**) represent the normalized values obtained by dividing the data in (**c**) by the data using the anti-N-domain antibody in (**b**) ± error propagated by the calculations. The data represents mean ± s.d. for $n = 12$ (-ATP/Mg2+) or 10 (+ATP/Mg2+) images from two independent experiments (**e**) and mean ± s.e.m. for three independent experiments (**f**). Source data are provided as a Source Data file.

fluorescence traces that showed single donor and single acceptor photobleaching events (Fig. 4b and Supplementary Fig. 6a). At the mixing ratio we used, we obtained the majority of the single-molecule FRET traces from NSFs with two N-MT subunits, with minimal interference from hexamers with three or more N-MTs (but with only two N-MTs labeled) because of their reduced binding avidity for αSNAP-loaded SNARE complexes.

As a benchmark, we first prepared a WT counterpart by mixing unlabeled and SNAP-tag-labeled WT protomers at the same molar ratio and determined their FRET distribution. We could discern

three Gaussian distributions, in which the two higher-FRET populations showed significant overlap and together constituted 84% of the population (Fig. 4c). The low-FRET population centered around 0.19 was more distinct and accounted for 16% of all the 20S complexes. We attributed these low-FRET populations to the "diagonal" arrangement in which labeled subunits appear in diagonal positions and thus have the largest inter-subunit distance (Fig. 4a, Type AD in the 2 N-MT arrangements). The other two groups (Type AB and AC) likely account for the two higher-FRET peaks. We noted that the measured fractions of these peaks also

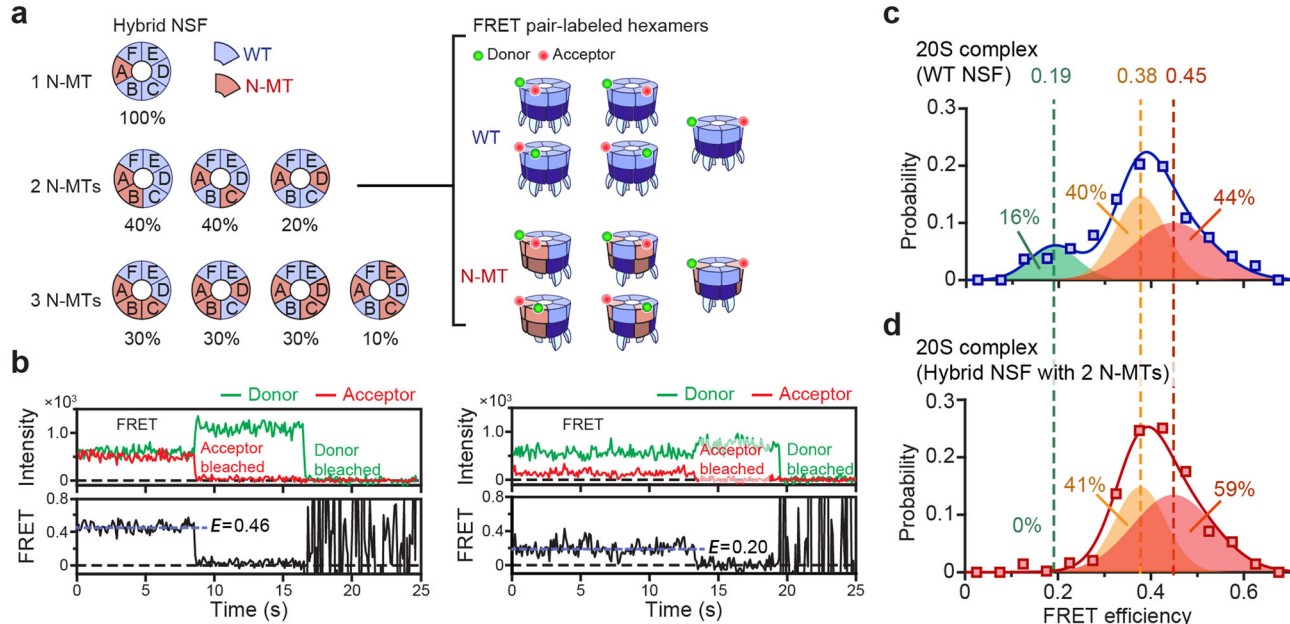

**Fig. 4 Identification of the core subunit connectivity within 20S complexes. a** The configuration (position) and the fraction of N-MT hybrid NSF, including those with one to three mutant subunit(s) with rotational symmetry. For hybrids including two N-MT, double-labeled hexamers of WT and N-MT hybrid NSF for single-molecule FRET experiments are depicted. **b** Representative fluorescence intensity with only one donor and acceptor bleaching and FRET efficiency traces at FRET efficiency values 0.46 and 0.20. **c, d** Distributions of FRET efficiency in double-labeled WT ($N = 765$ molecules) (**c**) and N-MT hybrid ($N = 491$ molecules) (**d**) NSF hexamer pull-downs via the SNARE-αSNAP complex. Best-fit 1D Gaussian mixture models are shown with each corresponding proportion. Source data are provided as a Source Data file.

roughly matched the calculated probabilities for randomly assembled hexamers, supporting our assignment (Fig. 4a vs. c).

Strikingly, when examining the FRET values sampled by the N-MT hybrid hexamers, we found a selective loss in the low-FRET population (Fig. 4d). The middle population was also reduced slightly in the N-MT preparation in comparison with the highest-FRET peak, but this change was relatively minor. This result crucially indicates that the binding capacity of an NSF hexamer is severely impaired if two diagonal N-domains are simultaneously removed.

We next asked whether the "diagonal" rule obtained from the single-molecule FRET experiments can explain our earlier results with N-MT hybrid mixtures (Fig. 3d). In the case of the double-mutant, the rejection of Type AD is consistent with our observation that the relative avidity of arrangements containing two N-MT subunits is reduced by 20% compared with those carrying one N-MT subunit (Fig. 3d vs. 4a). In arrangements with three N-MT subunits, we noted that Type ABD and ACD should be rejected by the diagonal rule (Fig. 4a). Although Type ACE does not contain two diagonally located N-MTs, the published EM structures make it unlikely that all three N-domains in Type ACE bind an αSNAP dimer simultaneouly[24]. Type ACE must then use only two N-domains for 20S complex assembly (Supplementary Fig. 6b), a scenario at odds with our finding that the hybrid hexamers with only two N-domains cannot form 20S complexes (Fig. 3d). Although we could not rule out a minor 20S complex population carrying Type ACE (10% of hybrid NSF hexamers) and more than two αSNAPs, Type ABC is likely the main functional arrangement of the three N-MT hybrid hexamers capable of 20S complex assembly (Figs. 3d and 4a).

Our approach with single-protomer resolution suggests that three neighboring NSF subunits and an αSNAP dimer define the minimal unit for 20S complex formation. We observed remarkable cooperativity in their mechanical coupling such that any further reduction from this minimal set weakens the interaction of the remaining components, leading to total impairment in 20S

complex formation. At the same time, 20S complexes maintained by this minimal mechanical coupling are fully functional for SNARE complex disassembly. This high cooperativity in mechanical coupling and solidification would prevent the formation of any dysfunctional complexes that nonproductively consume ATP. This would allow cells to be parsimonious in the energy expenditure of SNARE complex disassembly (see "Discussion" for details).

**Walker B mutant differentially affects the two ATP hydrolysis modes of the 20S complex.** We next wondered whether the ATPase activity of NSF changes upon 20S complex assembly. We prepared another mutant protomer, A-MT, with an E329Q mutation in the Walker B motif of the D1 domain to impair its ATP hydrolysis activity[28,41,42]. Following the procedures established above, we generated A-MT hybrid hexamers at a molar ratio of WT:A-MT = 6:4 (mol/mol) (Fig. 5a and Supplementary Fig. 7a).

We first examined the 20S complex-forming capability of the A-MT hybrid hexamers (prepared). Under non-ATP-hydrolyzing conditions, the counts for 20S complex formation measured for WT and A-MT hybrid NSF hexamers were comparable to one another, as expected from A-MT's intact N-domains (Fig. 5b). Under ATP-hydrolyzing conditions, the WT NSF hexamers catalyzed SNARE complex disassembly and were completely dissociated from the surface vesicles. For the A-MT hybrid hexamers, however, the same reaction conditions did not induce any changes in the 20S complex counts, nor in those of the SNARE complexes (Fig. 5b and Supplementary Fig. 7b). This indicates that the presence of the A-MT subunits severely affected the disassembly activity of the 20S complex.

By studying the photobleaching steps of individual 20S complexes, we further scrutinized a scenario in which some mildly affected A-MT hybrid hexamers (thus retaining activities) were replaced by those without activities, shifting the internal

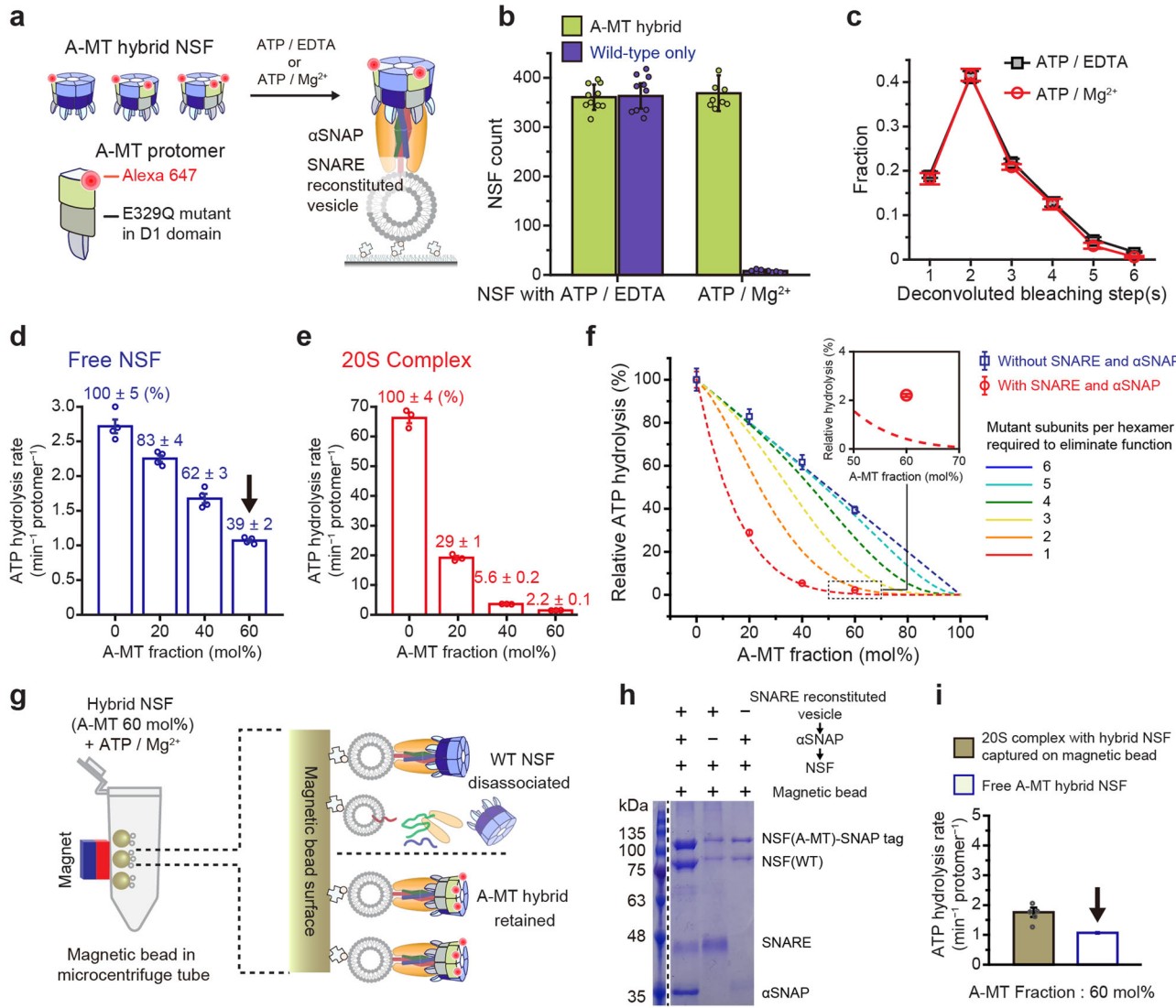

**Fig. 5 Walker B mutant differentially affects the two modes of NSF ATPase activity. a** Preparation of A-MT hybrid NSF that contain ATP-hydrolysis mutant protomer (E329Q). Experimental scheme for A-MT hybrid NSF binding to the SNARE–αSNAP complex. **b** Comparison of counts of A-MT hybrid NSF and wild-type NSF binding to the SNARE-αSNAP complex under ATP-hydrolyzing (1 mM ATP/10 mM Mg²⁺) and non-hydrolyzing (1 mM ATP/1 mM EDTA) conditions. **c** Distributions of the deconvoluted photobleaching step(s) for A-MT hybrid NSF at the two conditions in (**b**) considering 90% labeling efficiency. **d, e** Measurements of ATP hydrolysis rates without (**d**) or with (**e**) SNARE (300 nM) and αSNAP (1 μM) with varying A-MT mutant fractions. **f** Measured ATP hydrolysis rates in (**d**, **e**) normalized to WT samples (0% A-MT). Dashed lines indicate model curves assuming a particular number of mutant subunits per hexamer required to eliminate function. Inset is a magnified view of the dotted box. **g** Schematic for the removal of WT NSF hexamers in A-MT hybrid NSF samples using magnetic beads. Only NSF containing at least one A-MT subunit stably trapped on the magnetic beads with WT NSF hexamer removed by washing. **h** Representative SDS-PAGE gel image of proteins binding to magnetic beads under three different conditions after measuring the ATP hydrolysis rate. **i** Comparison of the measured ATP hydrolysis rate of NSF on the magnetic beads (WT removed) and that of free NSF with 60% A-MT fraction. The two arrows in (**d**, **i**) indicate the same data. Error bars in (**b**) represent mean ± s.d. for n = 10 for A-MT and n = 8 for Wild-type images from two independent experiments. Error bars represent mean ± s.e.m. from three independent experiments (**c**, **e**) and four independent experiments (**d**) and five independent experiments (**i**). Error bars in (**f**) represent normalized value ± propagated error by the calculations. The reproducibility of (**h**) was confirmed in two independent gel images. Source data are provided as a Source Data file.

distribution of the hybrid hexamers in one way or another. We found that the ATP-hydrolyzing condition did not alter the number or distribution of A-MT subunits over the entire stoichiometric range (Fig. 5c). In particular, we did not observe any reduction in single or double A-MT populations. Thus, the presence of even a single A-MT subunit appears to abolish the disassembly activity of the hybrid hexamer, suggesting a high level of coordination between the NSF subunits[42].

We next sought to determine the biochemical step in which the observed inter-subunit coordination takes place. In one scenario,

it is possible that the WT subunits in the A-MT hybrid hexamers can still hydrolyze ATP, but that their ATP hydrolysis does not lead to global conformational changes because of the neighboring A-MT subunits. In an alternative scenario, the ATP hydrolysis events are closely synchronized with another, and the presence of a single A-MT subunit prevents ATP hydrolysis of all other five WT subunits.

To address this question, we measured the ATPase activity of A-MT hybrid hexamers (Fig. 5d, e and Supplementary Fig. 7c, d). Without cofactors (αSNAP and SNARE complexes), the free

hybrid hexamers showed a low, basal rate of ATP hydrolysis (Fig. 5d and Supplementary Fig. 7c). Of note, increasing the fraction of A-MT reduced basal activity in a strictly proportional manner, indicating that these unstimulated hydrolysis events in the WT subunits occur independently of nearby A-MT subunits (Fig. 5f).

We next added soluble cofactors for the formation of the 20S complex and observed a 24-fold increase in the rate of ATP hydrolysis for the WT hexamers, consistent with previous reports (compare the first bars of Fig. 5d, e)[43]. Importantly, as we increased the A-MT fraction, stimulated ATP hydrolysis showed a steep reduction even with small amounts of A-MT (Fig. 5e). We found that this non-linear decrease in the ATP hydrolysis rate could only be explained by a model where the inclusion of a single A-MT subunit eliminated the stimulated ATP hydrolysis activity of the entire NSF hexamer (but not the basal activity, see below) (Fig. 5f). A slightly less stringent model that required two A-MT subunits to quench the stimulated ATP consumption failed to account for the observed steep decrease (Fig. 5f).

At the same time, we found that the stimulated hydrolysis rates in A-MT hybrid NSF were slightly but consistently higher than those predicted by the completely coupled scenario (Fig. 5f, inset). If a small fraction of WT homohexamers in A-MT hybrid samples are responsible for any observed residual ATPase activity, the fraction of pure WT homohexamers is predicted to be $(0.4)^6$ ~0.4% at the mixing ratio we used (60 mol% of A-MT hybrid). However, we found this estimation is noticeably smaller than the observed 2.2% activity relative to the purely WT sample (Fig. 5f, inset).

To determine the origin of the observed residual ATP hydrolysis, we designed a scheme to remove WT homohexamers from our A-MT hybrid preparations (Fig. 5g). We took advantage of our observation that NSF hexamers containing at least one A-MT subunit are stably trapped in 20S complexes (Fig. 5c). By assembling 20S complexes on magnetic beads under ATP-hydrolyzing conditions, we induced dissociation of WT NSFs while retaining A-MT-carrying hybrid hexamers on the bead surface (Fig. 5g, h and Supplementary Fig. 7e). When we measured ATP hydrolysis after thorough washing, we found that there was still residual ATPase activity (Fig. 5i and Supplementary Fig. 7f). In addition, the observed rate, normalized for each protomer, was almost equal to that of free NSFs at the same A-MT fraction (Fig. 5i). This crucially implies that slow, independent hydrolysis events continue to occur in the NSF hexamers forming the 20S complexes.

Together, our results reveal two distinct modes of ATP hydrolysis within 20S complexes. In the basal activity of unstimulated NSFs, all six subunits work independently of one another, binding and hydrolyzing ATPs largely at random (Fig. 5d, f). We found that this slow, random ATP hydrolysis persists in NSFs within 20S complexes (Fig. 5i). The second mode, specific to the 20S complexes, is much faster and highly coordinated, strongly linking ATP hydrolysis events in all six subunits (Fig. 5e). We found that inhibition of a single subunit abolishes this mode of stimulated ATP hydrolysis in the entire hexamer (Fig. 5f). Our results also indicate that this synchronized ATP hydrolysis is directly coupled to global conformational changes in NSF (Fig. 5b), probably by generating large molecular tension used in a "tug-of-war" with the substrate SNARE complex.

**Construction of the ATP hydrolysis cycle of the 20S complex.** Finally, we attempted to construct the ATP hydrolysis cycles for the 20S complex (Fig. 6a). We assumed a constant rate of $\gamma$ for basal ATP hydrolysis because we found this mode of ATP hydrolysis occurs in each ATP-bound subunit independent of other subunits. With $n$ of six subunits occupied with ATP, the total rate of basal ATP hydrolysis would simply be $n\gamma$. When the hexamers are fully loaded with ATP, stimulated hydrolysis occurs in which ATP hydrolysis in all six subunits is strongly coupled at a faster rate $\beta$ (Fig. 6a). In this model, the balance between coupled and independent hydrolysis dictates the degree of inter-subunit cooperativity.

To infer the nature of the transitions between different ATP-occupancy states, we measured total ATP hydrolysis activity in 20S complexes as a function of ATP concentration (Fig. 6b and Supplementary Fig. 8a). Contrary to what is suggested by the existence of a strongly coupled ATP hydrolysis mode, the total ATP hydrolysis rate does not exhibit a strong sigmoidal curve. When we fitted the curve with the Hill function, cooperativity was modest, yielding a Hill coefficient of $1.33 \pm 0.03$ (Fig. 6b). This suggested that the binding of ATP in one subunit does not accelerate (or deaccelerate) ATP binding in the other subunits. We, therefore, assumed that ATP binding to each subunit occurs independently and that the dissociation constants of ATP ($K_d$) from individual subunits are identical (Fig. 6a).

When we applied our constructed model to the ATP titration data, we found the stimulated hydrolysis rate ($\beta = 59 \pm 1 \text{ min}^{-1}$) to be an order of magnitude faster than the basal hydrolysis rate ($\gamma = 7 \pm 2 \text{ min}^{-1}$) (Fig. 6C). We estimated the $K_d$ to be $4.8 \pm 0.3$ μM, such that most of the ATP pockets would be filled under physiological conditions (see below). Our results suggest the stimulated ATP hydrolysis tightly linking all six subunits of NSF is sufficient to describe the observed cooperativity, without the need to invoke any cooperative ATP binding. In fact, the Monod–Wyman–Changeux (MWC) model[44], which is widely adopted to describe cooperative behaviors, was reduced to a simpler model in Fig. 6a when fitted to the measured values (Supplementary Fig. 8b, d).

Finally, we developed a separate model for the basal hydrolysis of free NSF hexamers (Fig. 6d and Supplementary Fig. 8f). Surprisingly, measurements of the basal activity at varying concentrations of ATP revealed negative cooperativity among NSF subunits (a Hill coefficient of $0.58 \pm 0.09$) (Fig. 6e). Consequently, the single-$K_d$ approach used for the stimulated mode did not produce a good fit for the ATP dependence (Supplementary Fig. 8g, h). We thus adopted the Koshland–Nemethy–Filmer (KNF) model[45] that introduces a scaling factor, $\alpha$, such that the $n$th ATP binding event becomes less favorable with an increasing dissociation constant of $\alpha^{n-1}K_d$ ($\alpha > 1$) (Fig. 6d). This model produced a successful fit for our data with best-fit parameters of: $\gamma = 2.9 \pm 0.2 \text{ min}^{-1}$, $K_d = 11 \pm 3$ μM, and $\alpha = 1.6 \pm 0.3$ (Fig. 6f). We attributed the small difference in the $\gamma$ values between free NSFs and 20S complexes to a small population of hollow 20S complexes devoid of SNAREs (Supplementary Fig. 8i). We applied the KNF model to the ATP titration curve of the 20S complexes in Fig. 6b, c, and found that the theoretical model converged once again to the simpler model in Fig. 6a with a scaling factor of nearly 1, assuring the validity of the ATP hydrolysis cycle we constructed for the 20S complexes (Supplementary Fig. 8c, e).

Using the ATP hydrolysis cycles constructed above, we estimated the ATP-binding equilibrium of NSF hexamers in different states. In free NSFs, the slightly increased $K_d$ and negative cooperativity keeps the number of ATP molecules per hexamer considerably smaller than those in 20S complexes (Fig. 6g). Together, our results suggest 20S complexes take advantage of two ATP hydrolysis modes at opposite extremes in terms of biochemical cooperativity (one effectively random and the other totally coupled). This allows cells to concentrate their energy expenditure into functional 20S complexes while minimizing the number of ATP molecules consumed by free, idle NSFs.

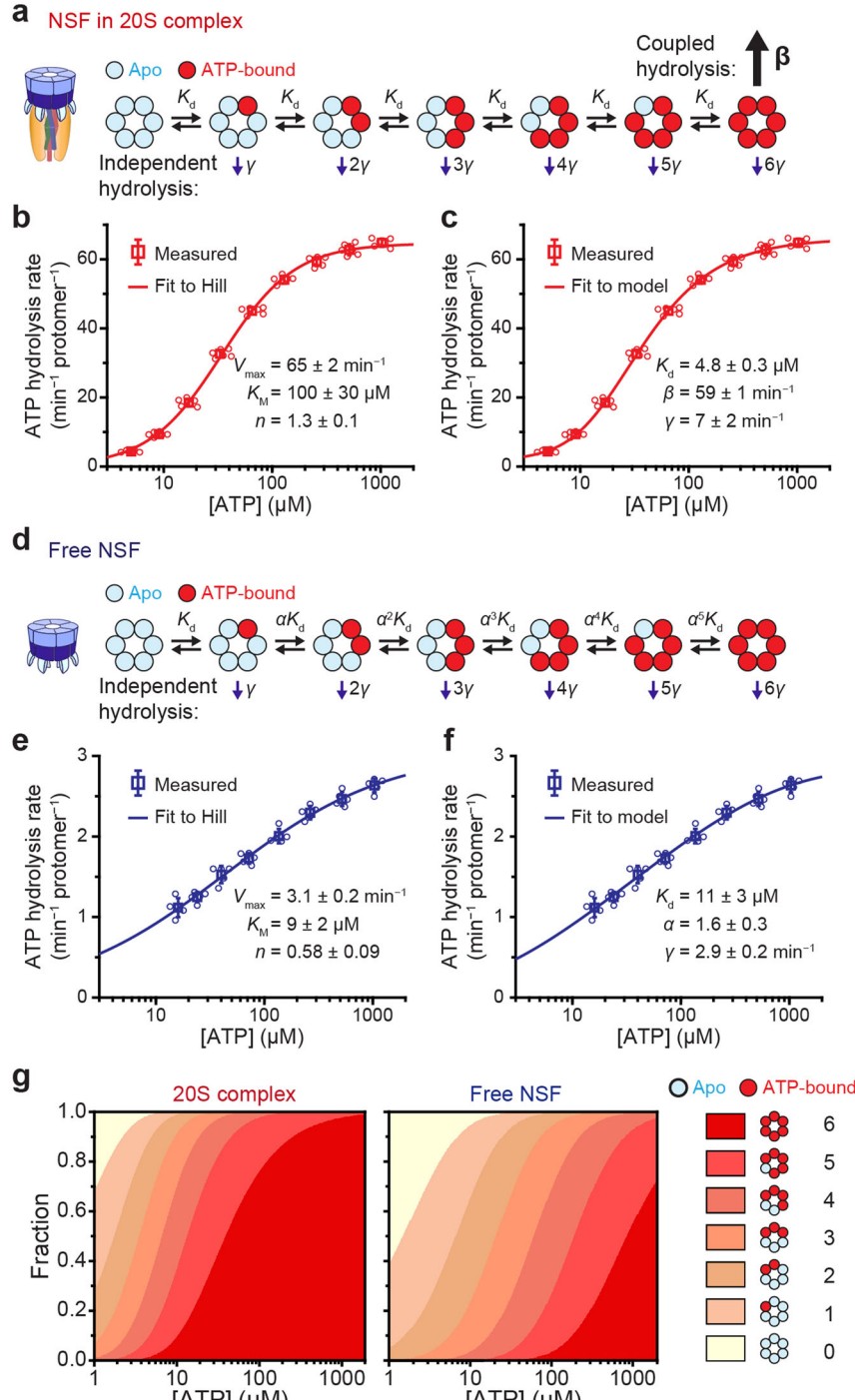

**Fig. 6 Construction of the ATP hydrolysis model for the 20S complex. a–f** Models for ATP binding and hydrolysis of NSF in 20S complexes (**a**) and free NSF (**d**) with corresponding measurements of hydrolysis rates at varying ATP concentrations (**b, c**) and (**e, f**), respectively. The same data were fitted either with a Hill equation (**b, e**) or with the model (**c, f**) proposed in (**a, d**), respectively. $K_M$, $K_d$: dissociation constant for ATP binding in subunits; $\gamma$: independent hydrolysis rate; $\beta$: coupled hydrolysis rate; $\alpha$: scaling factor for $K_d$ in the basal model; $n$: Hill coefficient; $V_{max}$: maximum rate of ATP hydrolysis. For direct comparison, the two rates $\gamma$ and $\beta$ have been normalized per protomer and unit time (**b, c**). **g** The fraction of apo-state NSF hexamers to fully loaded hexamers in the 20S complex and to the free state, depending on ATP concentration. Error bars represent mean ± s.d. from six independent experiments for (**b, c**) and five independent experiments for (**e, f**). Source data are provided as a Source Data file.

## Discussion

In this study, by employing an array of single-molecule force and fluorescence spectroscopy techniques, we found that the 20S complex has developed two sophisticated layers of proofreading in its ATP expenditure. Our single-molecule magnetic tweezer data suggested that the 20S complex induces efficient and

simultaneous disassembly of all four SNARE motifs in the SNARE complex on millisecond timescales. With the improved temporal resolution of our magnetic tweezers, we observed that 20S complex-mediated disassembly exhibits a unique unfolding intermediate centered around the +4 layer ($I_{20S}$) (Fig. 1i and Supplementary Fig. 2c), the intermediate not observed in the

mechanical pulling of lone single SNARE complexes. Notably, surface-exposed Syx and VAMP2 residues around the +4 and +5 layers work as the main binding site for αSNAPs in the C-terminal half of the SNARE complex[23,25]. There is another αSNAP-binding site in the −2 layer (composed mainly of the residues of SNAP-25) that presumably induces further unraveling of the SNARE motifs in the N-terminal half[25,37]. Thus, our observations suggest that the 20S complex induces highly directional disassembly of single SNARE complexes, during which the $I_{20s}$ state defines a transformational point where the 20S complex changes the point of the force application.

Using a single-molecule fluorescence technique to achieve single-protomer resolution, we next examined subunit connectivity at the interface between αSNAPs and NSF. We found that three neighboring NSF N-domains and an αSNAP homodimer define the minimal, core components that connect NSF and αSNAPs. We found remarkable cooperativity in the formation and consolidation of mechanical coupling between NSF and αSNAPs, indicative of the same level of selectivity in NSF's grabbing of the SNARE complex substrate. It may be envisaged that this mechanical selectivity confers advantages for cell energy expenditure by preventing the formation of aberrant 20S complexes that do not make the reliable mechanical connections required for SNARE complex disassembly (e.g., those connected using a single copy of SNAP, only two N-domains, or three N-domains with vacancies between them). Such dysfunctional 20S complexes, with slippery grips on substrate SNARE complexes, would simply consume ATP molecules without productive disassembly events. Thus, as a global remodeler that must overcome exceedingly large free energy barriers in a single step, the 20S complexes might have evolved to allow for only the productive complexes, a unique strategy not found with processive linear and rotational translocases[30,31,46]. We further note that according to the previous EM structures, the core mechanical coupling is critically disrupted concurrent with the disassembly event[12,13,23]. This may explain how the rigid 20S complex structure is so quickly dismantled upon disassembly of its target SNARE complexes.

We also found a high level of cooperativity in NSF's ATP consumption, another remarkable way the 20S complex is adapted for its mechanical function. Strikingly, we observed that NSFs in the 20S complex consume ATP in two distinct ways: one totally random and the other totally synchronized. After grabbing its substrate SNARE complex, NSF hexamer subunits show random ATP binding and hydrolysis to load itself with ATP molecules. Once fully charged, NSF synchronizes all six subunits to induce ATP hydrolysis at a rate increased by one order of magnitude. ClpX links two or three subunits (maximum four) in its burst ATP hydrolysis[46–48], a lower level of synchronization that allegedly gives ClpX more flexibility in its pulling to cope with the massive diversity in its substrates across the proteome[49]. In contrast, NSF seems to have evolved to specialize in the disassembly of one type of very rigid substrate—the SNARE complex —and has therefore developed the highest possible level of synchronization[50].

Why has NSF adopted random ATP binding and hydrolysis rather than cooperative ATP binding in reloading itself with ATPs? The viral DNA-packaging motor φ29 shows cooperative and ordered binding of ATPs in its homopentameric ring structure but with an irreversible biochemical step separating each ATP-binding event[31,51,52]. This defies the hypothesis that molecular motors have not evolved enough through sequence space to reach such perfect coordination. The answer may lie in the fact that NSF and the processive linear/rotational translocases have drastically different modes of mechanical action. Cooperative ATP binding quickly fills vacant subunits, serving well the linear/

rotational translocases that go through many consecutive mechanochemical cycles along their substrates[31,32,53]. In contrast, a similar cooperative ATP binding mechanism would lead to a lot of unnecessary ATP binding in NSF, which spends considerable time in an idle state in between bindings with the next SNARE complex substrate.

In free NSFs mimicking such an idle state, we indeed observed unexpected negative cooperativity in ATP binding. We measured the dissociation constant of the sixth ATP binding and found it to be ten times higher than that of the first ATP binding. This would significantly reduce the number of ATP molecules bound to free NSFs. Upon 20S complex formation, NSF lifts its inter-subunit inhibition and switches to zero cooperativity in ATP binding (i.e., total random binding), which effectively increases the average ATP occupancy of NSF hexamers. Thus, by combining two completely different modes of ATP hydrolysis, NSF achieves highly efficient mechanical function while minimizing ATP consumption in the idle states as it enters between global disassembly events.

The phylogeny suggests NSF is uniquely positioned in the entire AAA + family, with the p97s in mammalian cells being its closest homologs[54]. Cryo-EM structures indicate p97 and NSF show commonality in domain architecture and in the large conformational changes triggered by ATP hydrolysis[55,56]. In particular, p97 also exploits a diverse array of adaptors in binding to its substrates[57,58]. We expect that many features of the strategies NSF has developed for SNARE complex disassembly are shared by p97 as it handles its diverse list of substrates to maintain homeostasis of the human proteome.

## Methods

**Protein constructs.** Chinese hamster NSF was cloned into pET28a with an N-terminal His-tag and a thrombin cleavage site for protein purification. For the fluorescence assays, a SNAP tag was fused to the C-terminal end of NSF with a GGSG linker (in this case, the His-tag was moved from the N-terminus of NSF to the C-terminus of the SNAP tag). For N-domain-deleted NSF (N-MT), the Pre-Scission cleavage sequence (LEVLFQGP) was inserted in exchange for four residues between K202 and S207 of WT NSF. Bovine αSNAP was also cloned into pET28a with an N-terminal His-tag and a thrombin cleavage site. For single-molecule fluorescence assays, a SNAP tag was fused to the N-terminus of αSNAP with a GGSGGGSG linker and the His-tag was moved to the C-terminal end of αSNAP. All SNARE proteins including a truncated version of syntaxin-1A without the N-terminal $H_{abc}$ domain, synaptobrevin-2/VAMP2, and cysteine-free variant of SNAP-25 isoform A were derived from rats. For magnetic tweezer experiments, N-terminal His-tagged cysteine-free SNAP-25A, synaptobrevin-2 (2-97, L32C/I97C), and syntaxin-1A (191-268, I202C/K266C) were cloned into pET28a vector. Among four cysteines, two cysteines (I97C in synaptobrevin-2, K277C in syntaxin-1A) were used to conjugate the DNA handles for tweezing. The other two cysteines were used to form a disulfide bond for repeated disassembly and reassembly. For vesicle reconstitution of the ΔN-SNARE complex, syntaxin-1A without the $H_{abc}$ domain (183–288) and a synaptobrevin-2 fragment (49–96) were cloned into pETDuet-1 and co-expressed with an N-terminal His-tagged full-length SNAP-25 cloned into pET28a. For the fluorescence assays, synaptobrevin-2 lacking a transmembrane domain (1–96, A82C) was cloned into pGEX-KG with an N-terminal GST-tag and a thrombin cleavage site for protein purification. For the ATPase assays, SNARE complexes without transmembrane domains assembled from syntaxin-1A (183–262), synaptobrevin-2 (1–96), and full-length SNAP-25 were used. Syntaxin-1A (183–262) and synaptobrevin-2 (1–96) were cloned into pGEX-KG with N-terminal GST-tags and thrombin cleavage sites. SNAP-25 was cloned into pET28a with an N-terminal His-tag. Except for the SNAP-tag derivatives, the remaining constructs were used as previously described[26].

**Protein expression and purification.** Wild-type NSF and NSF-SNAP were expressed overnight at 37 °C in 15 mL of standard LB medium and additionally expressed overnight at 25 °C after being transferred to 1.5-L autoinduction LB medium. The cells were collected by centrifugation and the cell pellets were suspended in Buffer A (50 mM pH 7.4 HEPES, 300 mM NaCl, 20 mM imidazole, 0.5 mM TCEP). 0.5% Triton X-100 and 1× protease inhibitor cocktail (genDEPOT) were added to the suspended cells and the cells were lysed by sonication. After centrifugation at 17,000×$g$ for 30 min, the supernatant was nutated with Ni-NTA agarose resin (QIAGEN) for ~1 h at 4 °C. The resin was washed with Buffer A and the proteins were eluted with elution buffer (50 mM pH 7.4 HEPES, 200 mM NaCl, 400 mM imidazole, and 1 mM DTT). Thrombin for cleaving the His-tag and 1 mM

ATP, 1 mM EDTA, and 10% glycerol was added to the eluate. The eluate was concentrated and loaded onto a Superdex 200 16/600 column (GE Healthcare) pre-equilibrated with NSF buffer (50 mM pH 7.4 HEPES, 150 mM NaCl, 1 mM DTT, 1 mM ATP, 1 mM EDTA, and 10% glycerol). Only the hexameric fractions were concentrated again and loaded onto the same type of column pre-equilibrated with monomerization buffer (50 mM pH 8.0 sodium phosphate, 150 mM NaCl, and 1 mM DTT) to remove ATP. The same process was repeated again for monomerization. The monomeric fraction was concentrated to the desired concentration, frozen at −80 °C, and stored for future use.

The other proteins were grown in 1 L of LB medium at 37 °C until the $OD_{600}$ reached ~0.7. Expression was induced with the addition of 0.5 mM IPTG. The same process was followed for αSNAP, and SNAP-tag-αSNAP until imidazole elution. The eluate was concentrated and loaded onto a Superdex 200 10/300 increase column (GE Healthcare) pre-equilibrated with SEC buffer (50 mM pH 7.4 HEPES, 50 mM NaCl, and 1 mM DTT). αSNAP and its derivatives were freshly purified before use and never frozen.

The same process was followed for full-length SNAP-25A, synaptobrevin-2 (2-97, L32C/I97C), and syntaxin-1A (191-268, I202C/K266C) as for αSNAP. For magnetic tweezer experiments, the three SNARE components were mixed at an equal molar ratio and incubated for 3 h at room temperature for assembly. Assembled SNARE complexes were dialyzed with thrombin against PBS containing 1 mM DTT overnight at 4 °C to remove His-tags. To remove DTT, SNARE complexes were desalted using PD MiniTrap G25 column (GE Healthcare), and these desalted SNARE complexes were used to conjugate DNA handles.

The same purification process was followed for the soluble part of syntaxin-1A (183–262) and synaptobrevin-2 (1–96, WT and A82C), except that Glutathione-Sepharose 4B (GE healthcare) was used instead of Ni-NTA resin. The resin was sequentially washed with 0.1% Triton X-100-containing wash buffer (50 mM pH 7.4 HEPES, 300 mM NaCl, and 0.5 mM TCEP). The protein was then eluted by cleavage with thrombin. The eluate was concentrated and immediately loaded onto a Superdex 200 10/300 increase column pre-equilibrated with SEC buffer (50 mM pH 7.4 HEPES, 150 mM NaCl, and 0.5 mM TCEP). The purified synaptobrevin-2 (A82C) was used for labeling and the three SNARE components (SNAP-25, syntaxin-1A, synaptobrevin-2) were mixed in a 1:1:1 molar ratio for 30 min at 37 °C. The concentrated SNARE complex was then loaded onto a Superdex 200 16/600 column and stored at −80 °C.

Cells expressing the ΔN-SNARE complex (SNAP-25, syntaxin-1A (183–288), and a synaptobrevin-2 fragment (49–96)) were collected by centrifugation and suspended in Buffer A. 0.5% Triton X-100, 0.5% sarcosine, and a 1× protease inhibitor cocktail (genDEPOT) were added to the suspended cells, and the cells were lysed by sonication. The lysate was nutated for 1 h at 4 °C to achieve full solubilization of the membrane proteins. After centrifugation (17,000×g, 30 minutes), the supernatant was loaded onto Ni-NTA resin and nutated for 1 h at 4 °C. The resin was then sequentially washed with Buffer A containing 0.1% Triton X-100 and Buffer A containing 1% n-octyl-β-D-glucopyranoside (OG) before being eluted with elution buffer (50 mM pH 8.0 Tris-HCl, 20 mM NaCl, 1% OG, and 1 mM DTT). The eluate was loaded onto a HiTrap Q (GE Healthcare) for anion exchange chromatography with a linear gradient of 20–1000 mM NaCl in 50 mM pH 8.0 Tris-HCl, 1 mM DTT, and 10% glycerol. The purified ΔN-SNARE complexes were stored at −80 °C.

All purified proteins were confirmed by SDS-PAGE electrophoresis. Protein concentrations were measured with the NanoDrop2000C (Thermo Scientific) or with Bradford protein assays (Bio-Rad).

**Protein labeling and NSF re-hexamerization.** Purified soluble synaptobrevin-2 with a cysteine mutation (A82C) was labeled with ten times excess Cy3-maleimide (Lumiprobe) overnight at 4 °C. The free dyes were separated on a PD MiniTrap G25 column (GE Healthcare) pre-equilibrated with protein buffer (50 mM pH 7.4 HEPES, 150 mM NaCl, 10% glycerol, and 1 mM DTT). Labeling efficiency was measured using a NanoDrop2000C (Thermo Scientific) and the labeled protein was stored at −80 °C.

Purified SNAP-tag-αSNAP was diluted to 5 μM and labeled with 10 μM benzylguanine (BG)-Alexa647 dyes (New England Biolabs) for 2 h at room temperature or overnight at 4 °C. The free dyes were also separated on a PD MiniTrap G25 column pre-equilibrated with protein buffer (50 mM pH 7.4 HEPES, 150 mM NaCl, and 1 mM DTT).

In total, 5 μM monomeric NSF-SNAP tag was also labeled with BG-Alexa647 or BG-DY549 (New England Biolabs) overnight at 4 °C. To make hybrid NSF hexamers, labeled NSF-SNAP-tag monomers and NSF-WT monomers were mixed in the desired ratio with 1 mM ATP and 1 mM EDTA at 4 °C for 2 h. For the N-domain deletion hybrids, PreScission protease was added for an additional 2 h. Re-assembled NSF loaded onto a Superdex 200 16/600 column with reassembly buffer (50 mM pH 7.4 HEPES, 50 mM NaCl, 1 mM ATP, 1 mM EDTA, 10% glycerol, and 1 mM DTT). Unlabeled free dyes were separated during this step. Hexameric fractions were stored at −80 °C.

**Magnetic tweezers experiments.** Mechanical unfolding of SNARE complexes was performed with a custom magnetic tweezers setup on an inverted microscope as previously described[33]. Single SNARE complexes were conjugated to two 500-bp dsDNA handles that were, in turn, attached either to a glass surface or to superparamagnetic beads (Dynabeads M-270, Invitrogen). The magnetic beads were manipulated by moving a pair of permanent magnets on a motorized stage placed 2–20 mm above the sample and their coordinates were tracked in real-time relative to the position of a polystyrene reference bead affixed to the glass surface. The magnetic force generated by the bead was pre-calibrated using the thermal fluctuations of the bead's location as a function of magnet distance. For each molecular construct, the integrity of the DNA handles was verified by checking the force-extension curve, and the presence of a single SNARE complex was checked using the signature un-/re-zipping and un-/refolding events in characteristic force regimes. For enzymatic disassembly of 20S complexes, 1 μM αSNAP and 1 μM wild-type NSF hexamers were sequentially injected into the sample chamber using a syringe pump. All measurements were performed in the presence of 2.5 μM SNAP-25 in 50 mM HEPES (pH 7.4) containing 50 mM NaCl, 10 mM $MgCl_2$, and 2 mM ATP supplemented with an ATP regeneration system (phosphocreatine and creatine kinase). To determine the lifetime of the intermediate state for 20S complex-mediated disassembly, high-speed traces were subjected to hidden Markov modeling and the period between entering and exiting the intermediate state expected from the Baum–Welch algorithm was calculated.

**Vesicle reconstitution.** All lipids were purchased from Avanti. A mixture composed of 44% 1-palmitoyl-2-oleoyl-glycero-3-phosphocholine (POPC), 25% cholesterol, 12% 1,2-dioleoyl-sn-glycero-3-phospho-L-serine (DOPS), 15% 1,2-dioleoyl-sn-glycero-3-phosphoethanoleamine (DOPE), 3% brain L-α-phosphatidylinositol-4,5-bisphosphate (PI(4,5)P2), and 0.7% 1,2-dioleoyl-sn-glycero-3-phosphoethanolamine-N-(cap biotinyl) (Biotinyl CapDPPE) was dried via vacuum pumping overnight. Dried lipids were solubilized with 3% OG-containing buffer (50 mM pH 7.4 HEPES and 150 mM NaCl) to make a 15 mM lipid solution. This was gently mixed for 1 h at 4 °C to ensure complete solubilization. This lipid solution and purified ΔN-SNARE complex (SNAP-25, syntaxin-1A (183–288), synaptobrevin-2 fragment (49–96)) were mixed at the desired protein to lipid ratio (1:500 for measuring NSF signal, 1:10,000 for the SNARE disassembly experiment). The final mixture consisted of 2% OG and 3 mM lipids. To reconstitute vesicles, this mixture was diluted three-fold with buffer (50 mM pH 7.4 HEPES and 150 mM NaCl) to lower the OG concentration below the critical micelle concentration (CMC). The residual OG in this diluted mixture was removed by dialysis at 4 °C overnight with SM2 bio-beads (Bio-Rad).

**Single-molecule fluorescence assay of NSF-mediated disassembly of SNARE complexes.** We used a quartz slide coated with a 40:1 molar ratio of mPEG and biotin-PEG (LaySan Bio) for total internal reflection fluorescence microscopy. All samples were diluted with H50N50 buffer (50 mM pH 7.4 HEPES, 50 mM NaCl). For the single-molecule SNARE disassembly assay, 0.1 mg/ml Neutravidin (Thermo Fisher Scientific) was incubated first on a quartz glass slide. After incubation for 3 min and washing with H50N50 buffer, 10 μM ([Lipid]) of either 1:10,000 (protein/lipid ratio) proteoliposome integrated with ΔN-SNARE complex or protein-free liposomes (negative control) was incubated for 3 min. After vesicle immobilization and a washing step to remove non-immobilized vesicles, 2 nM Cy3-labeled soluble synaptobrevin-2 was added and further incubated for 3 min. After another washing step, 10 μM αSNAP was incubated for 5 min before the αSNAP was removed by washing. For the formation of 20S complexes, 200 nM wild-type NSF, or 400 nM N-MT hybrid NSF with 1 mM ATP/1 mM EDTA was added and incubated for 3 min. To compare disassembly activity, 1 mM ATP/10 mM $MgCl_2$ was injected to wash out unbound NSF and to simultaneously induce disassembly of the complex. The reaction was stopped by washing with H50N50 buffer after 5 min for full disassembly or after several time points for measuring disassembly rate. To count Cy3-labeled synaptobrevin-2 spots, TIR fluorescence images were recorded with an EM-CCD camera (iXON DU897D, Andor) at a frame rate of 10 Hz for 2 s under illumination with a 532-nm laser. Spots were detected as local Gaussian maxima in the snapshot images.

**Single-molecule hybrid NSF assay and single-molecule FRET.** We used the same PEG-coated quartz slide used for the SNARE disassembly assay. All samples were diluted with H50N50 buffer containing 10 mg/mL BSA, and the washing buffer contained H50N50 buffer alone. In total, 0.1 mg/ml NeutrAvidin was added first to a quartz glass slide. After removing the free NeutrAvidin by washing, the slide was passivated with H50N50 buffer containing 10 mg/mL BSA for 5 min. After passivation, 100 μM ([Lipid]) of either a 1:500 (protein/lipid ratio) proteoliposome integrated with ΔN-SNARE complex or liposomes without protein as a negative control was added and incubated for 5 min. After vesicle immobilization and removal of the free vesicles by washing, 10 μM non-labeled soluble synaptobrevin-2 was added and incubated for 5 min. After removal of the free synaptobrevin-2 by washing, 100 nM αSNAP was added and incubated for 5 min. Then the unbound αSNAP molecules were removed by washing. The mixing experiments with wild-type and mutant αSNAP used the same total concentration, but with different proportions of the wild-type protein.

In total, 5 pM NSF (WT or A-MT hybrid) or 10 pM NSF (N-MT hybrid) with 1 mM ATP/1 mM EDTA (20 S complex loading) or 1 mM ATP/10 mM $MgCl_2$ (disassembly) was added and incubated for 5 min so that the number of stable 20S complexes could be compared. After washing with H50N50 buffer containing

1 mM ATP/1 mM EDTA, an oxygen scavenger system (2.5 mM PCA, 0.5 unit/ml PCD, 1 mM Trolox) with 1 mM ATP/1 mM EDTA was injected onto the slide for stable fluorescence time trace observations. Fluorescence time traces for counting photobleaching steps and for FRET were collected at a frame rate of 10 Hz for about 400 s. The number of bleaching steps was counted using a step-finding algorithm in MATLAB and double-checked by confirming the results with a curve fit.

For the single-molecule FRET assay, alternating laser exposures were used to find single donor/acceptor pairs. After short-term acceptor excitation to specify the molecule's position, subsequent donor excitations and red excitations were repeated to achieve donor and acceptor bleaching. The resulting FRET data were collected for NSF samples with only one donor and one acceptor, as discriminated by the bleaching step. FRET efficiency $E$ was calculated according to the formula $E = (I_A - l \cdot I_D)/(I_A + \gamma \cdot I_D)$, where $l$ is leakage and $\gamma$ is gamma factor, and $I_A$ and $I_D$ are donor and acceptor intensities corrected against the background signal molecule by molecule.

**Single-molecule αSNAP and NSF co-localization experiment**. This experiment was almost identical to the hybrid NSF assay. All samples were diluted in H50N50 buffer containing 10 mg/mL BSA and washed in pure H50N50 buffer. First, 0.1 mg/ml NeutrAvidin was added and incubated on a quartz glass slide. After removing the free NeutrAvidin by washing, the slide was passivated with H50N50 buffer containing 10 mg/mL BSA for 5 min. After passivation, 100 μM ([Lipid]) of either 1:30,000 (protein/lipid ratio) proteoliposome integrated with ΔN-SNARE complex or liposome without protein as a negative control was added and incubated for 5 min. After vesicle immobilization and removal of free vesicles by washing, 10 μM non-labeled soluble VAMP2 and 100 nM SNAP-tag-αSNAP were sequentially incubated for 5 min and washed out. Finally, 1 nM NSF labeled with Dylight549 with 1 mM ATP and 1 mM EDTA for co-localization was incubated for 5 min and washed out. For obtaining a stable fluorescence time trace, an oxygen scavenger system (2.5 mM PCA, 0.5 unit/ml PCD, 1 mM Trolox) was injected onto the slide. Fluorescence time traces were collected at a frame rate of 10 Hz for about 400 s so that photobleaching steps could be counted.

**Single-molecule immunoprecipitation assay for labeled hybrid NSF**. The same PEG-coated quartz slides were used for the single-molecule IP assays. All samples were diluted in and washed with H50N50 buffer. In total, 0.1 mg/mL NeutrAvidin was added to and incubated first on a quartz slide for 3 min before being removed by washing with H50N50 buffer. In all, 5 μg/mL secondary antibody (i.e., biotinylated goat anti-mouse or rabbit IgG antibody) and 5 μg/mL primary antibody (i.e., mouse monoclonal Anti-NSF (Santa Cruz Biotechnology) or rabbit polyclonal Anti-SNAP tag (New England Biolabs)) were added and incubated for 5 min. After removing the free antibodies by washing, 1 nM NSF for anti-N-domain or 20 pM NSF for anti-SNAP tag with 1 mM ATP and 1 mM EDTA was added and incubated for 5 min. The free proteins were removed by washing with H50N50 buffer containing 1 mM ATP/1 mM EDTA. An oxygen scavenger system (2.5 mM PCA, 0.5 unit/ml PCD, 1 mM Trolox) with 1 mM ATP/1 mM EDTA was injected onto the slide, and fluorescence time traces were collected at a frame rate of 10 Hz for about 400 s.

**Data processing of bleaching histograms**. To confirm SNAP-tag labeling efficiency, we made SNAP-tag dimers linked with GGSGGSG flexible linker with a C-terminal His-tag. The purification and labeling process was the same as that used for SNAP-tag-αSNAP, except that 2.5 μM of protein was used in the labeling reaction because of the dimeric nature. After removing the free dye using a PD MiniTrap G25 column, 10 pM SNAP-tag dimer diluted in H50N50 buffer was pulled-down with the single-molecule IP system using an anti-SNAP-tag antibody. We measured the ratio of traces with one and two photobleaching steps. Labeling efficiency was calculated by assuming a stochastic process. After measuring a ~90% efficiency, All bleaching histograms were deconvoluted according to the following equation: $R_i = A^{-1} \cdot X_i$, where $R_i$ is the expected real distribution with $i$ number of labeled subunits, and $X_i$ is the observed histogram with i number of photobleaching steps. $A^{-1}$ is the inverse of the $6 \times 6$ upper triangular matrix A with the following entries: $a_{ij} = {}_j C_i \cdot r^i (1-r)^{j-i}$ for $j \geq i$, otherwise 0.

**ATPase activity assay**. The ATPase activity of NSF was measured using an ATPase ELIPA Biochem Kit (Cytoskeleton, Inc.). A spectrophotometer was used to measure UV absorbance at 360 nm (± 5 nm bandwidth). All assays were carried out at 37 °C, and the signal was read every 30 s. Stimulated ATPase activity of NSF was measured in combination with 300 nM soluble SNARE complex, 1 μM αSNAP, and 3–8 nM NSF hexamer in ATPase reaction buffer (50 mM pH 7.4 HEPES, 50 mM NaCl, 1 mM ATP, and 5 mM MgCl₂) with the 2-amino-6-mercapto-7-methyl-purine riboside (MESG) and purine nucleoside phosphorylase (PNP) provided with the kit. An ATP titration assay was carried out under the same conditions, except for using a variable ATP concentration. NSF basal ATPase activity was measured under the same conditions, except for 30 nM NSF was used to obtain a stable signal. A phosphate buffer standard was measured in each experiment to calibrate the UV absorbance signal to the amount of inorganic phosphate. To

measure the initial ATPase rate, the first data points of the linearly increasing interval were used for linear regression analysis.

**NSF ATPase activity assay on magnetic beads**. This assay used streptavidin-coated magnetic beads (Invitrogen). In total, 30 μl of 1 mg/ml streptavidin-coated magnetic beads were transferred to a 1.5-ml tube. The magnetic beads were attached to the tube wall using a neodymium magnet and washed with H50N50 buffer. In all, 300 μl of 1 mM ([Lipid]) 1:500 (protein/lipid ratio) biotinylated proteoliposomes with ΔN complex were added to the tube containing the magnetic beads. After the reaction was allowed to proceed for 5 min, the magnetic beads were washed again. In all, 300 μl of 2 μM soluble synaptobrevin-2 and 800 μl of 1.5 μM αSNAP diluted in H50N50 buffer were sequentially added and incubated for 5 minutes before being removed by washing. In total, 100 μl of 200 nM NSF (60% A-MT subunit) diluted in H50N50 buffer containing 1 mM ATP/5 mM MgCl₂ was added and incubated for 1 min. After loading the NSF, H50N50 buffer containing 1 mM ATP/5 mM MgCl₂ was added and incubated for 3 min at 37 °C before being removed by washing. This was done to eliminate WT NSF hexamers completely. This process was repeated three times to ensure complete removal. Finally, 40 μl of H50N50 buffer containing 1 mM ATP/ 5 mM MgCl₂ was added to the tube and incubated for 10, 20, or 30 min at 37 °C. To measure the ATPase activity of stably trapped A-MT hybrid NSF, buffer containing inorganic phosphate ions were extracted from the tube while the beads were attached to the tube wall using a magnet. The amount of inorganic phosphate ion was measured in the buffer using an ELIPA kit (Cytoskeleton, Inc.). After all reactions were complete, 20 μl of SDS-PAGE gel loading dye was added to the tube. The protein solution containing the loading dye was then extracted to measure the amount of NSF trapped with the magnetic beads. The amount of trapped NSF was measured by running a PAGE gel with a known amount of NSF protein. The ATPase rate for the 20 S complex was calculated using the slope of the inorganic phosphate ion curve as measured at several time points and divided by the amount of NSF protein obtained from the gel.

**Models for ATP binding and hydrolysis in NSF**. The theoretical lines in Fig. 5f are following the equations below:

$$y = (1-x) \sum_{i=0}^{j-1} \binom{5}{i} (1-x)^{5-i} x^i \tag{1}$$

where $x$ is mutant fraction and $j$ is mutant subunits per hexamer required to eliminate function.

The OriginPro program was used for curve fitting for all the fitting results shown in Fig. 6 and Supplementary Fig. 8. All models for NSF assume that all ATP binding for NSF subunits occurs faster than ATP hydrolysis rate. That is, the NSF populations are pre-equilibrated and ATP hydrolysis does not affect the equilibrium population. Furthermore, we set the dimension of coupled ATP hydrolysis rate $\beta$ is per protomer and unit time for direct comparison with $\gamma$. However, the coupled ATP hydrolysis rate is also constant for hexamer because all ATPs are induced simultaneously. All functions used to fit the results appear below.

The Hill function used the following equation:

$$V = V_{max} \frac{[ATP]^n}{[ATP]^n + K_M^n} \tag{2}$$

where $V_{max}$: maximum rate of ATP hydrolysis; $K_M$: dissociation constant for ATP binding; $n$: Hill coefficient.

The function of the ATP hydrolysis model for NSF in the 20S complex used the following equation:

$$V = \frac{\beta \left(\frac{[ATP]}{K_d}\right)^6 + \gamma \left(\frac{[ATP]}{K_d}\right)\left(1 + \frac{[ATP]}{K_d}\right)^5}{\left(1 + \frac{[ATP]}{K_d}\right)^6} \tag{3}$$

where $\beta$: coupled ATP hydrolysis rate; $\gamma$: independent ATP hydrolysis rate; $K_d$: dissociation constant for ATP binding.

The function of the ATP hydrolysis model based on the MWC model for NSF in the 20S complex used the following equation:

$$V = \frac{\beta \left(\frac{[ATP]}{K_r}\right)^6 + \gamma \left(\frac{[ATP]}{K_r}\right)\left(1 + \frac{[ATP]}{K_r}\right)^5 + \gamma L \left(\frac{[ATP]}{K_t}\right)\left(1 + \frac{[ATP]}{K_t}\right)^5}{\left(1 + \frac{[ATP]}{K_r}\right)^6 + L \left(1 + \frac{[ATP]}{K_t}\right)^6} \tag{4}$$

where $\beta$: coupled ATP hydrolysis rate; $\gamma$: independent ATP hydrolysis rate per protomer; $L = \frac{T_0}{R_0}$ is the allosteric constant; $K_r$ and $K_t$: dissociation constant for ATP binding at $R$ form and $T$ form, respectively.

The function of the ATP hydrolysis model based on the KNF model for NSF in the 20S complex used the following equation:

$$V = \frac{\beta \left(\frac{[ATP]}{K_r}\right)^6 \alpha^{-15} + \gamma \left(\frac{[ATP]}{K_r}\right) \sum_{k=0}^{5} \binom{5}{k}\left(\frac{[ATP]}{K_r}\right)^k \alpha^{-\frac{k(k+1)}{2}}}{\sum_{k=0}^{6} \binom{6}{k}\left(\frac{[ATP]}{K_r}\right)^k \alpha^{-\frac{k(k-1)}{2}}} \tag{5}$$

where $\beta$: coupled ATP hydrolysis rate; $\gamma$: independent ATP hydrolysis rate; $K_d$: dissociation constant for ATP binding; $\alpha$: scaling factor.

The function of the ATP hydrolysis model for free NSF based on the KNF model used the following equation:

$$V = \frac{\gamma\left(\frac{[ATP]}{K_r}\right)\sum_{k=0}^{5}\binom{5}{k}\left(\frac{[ATP]}{K_r}\right)^k \alpha^{-\frac{k(k+1)}{2}}}{\sum_{k=0}^{6}\binom{6}{k}\left(\frac{[ATP]}{K_r}\right)^k \alpha^{-\frac{k(k-1)}{2}}} \tag{6}$$

where $\gamma$: independent ATP hydrolysis rate; $K_d$: dissociation constant for ATP binding; $\alpha$: scaling factor.

The function of the ATP hydrolysis model for free NSF with single $K_d$ used the following equation:

$$V = \frac{\gamma\left(\frac{[ATP]}{K_d}\right)}{\left(1 + \frac{[ATP]}{K_d}\right)} \tag{7}$$

where $\gamma$: independent ATP hydrolysis rate; $K_d$: dissociation constant for ATP binding.

**Reporting summary**. Further information on research design is available in the Nature Research Reporting Summary linked to this article.

## Data availability
Data supporting the findings of this manuscript are available from the corresponding author upon reasonable request. A reporting summary for this article is available as a Supplementary Information file.

## Code availability
Custom codes that support the findings of this study are available at a dedicated Github repository (https://github.com/tyyoonlab/Nature-Communications for the magnetic tweezer experiments; https://github.com/kahutia/SingleMoleculeImageAnalyzer/releases/tag/V8.0 and https://github.com/kahutia/smFRET_trace_viewer/releases/tag/v2.1 for the fluorescence experiments).

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

## Acknowledgements
This work was supported by a grant from the Bio & Medical Technology Development Program (NRF-2018M3A9E2023523) and the National Research Foundation of Korea funded by the Korean government (MSIT) (NRF-2020R1A5A1018081).

## Author contributions
T.-Y.Y. conceived the project. C.K. and T.-Y.Y. designed the experiments. C.K. purified all recombinant proteins. C.K. and M.J.S. performed all experiments and analyzed the data. S.H.K. provided analysis tools and advice regarding experimental design. G.S.E. performed data visualization and manuscript editing. J.-K.R. contributed the initial design of the work. C.H. and R.J. provided advice regarding manuscript review and editing. C.K., M.J.S., and T.-Y.Y. wrote the manuscript.

## Competing interests
The authors declare no competing interests.
