## [Peer Review File · Nature Communications]

Reviewer #1 (Remarks to the Author):

This beautiful paper describes an analysis of the mechanism of disassembly of the neuronal SNARE complex by NSF and α SNAP using single-molecule force measurements in combination with single-molecule fluorescence spectroscopy. Among multiple interesting results, the authors reveal an intermediate state in the disassembly pathway that reflects the initial step that leads to disassembly and likely involves opening of the C-terminus of the SNARE four-helix due to interactions with α SNAP. They also show that a minimum of two α SNAP molecules and three NSF N-terminal domains are required for disassembly, that such minimal 20S complexes are almost as efficient as WT 20S complex, and that the random nature of ATP binding to individual NSF subunits is converted to an almost perfect synchronization in ATP hydrolysis that results in the disassembly of the SNARE complex in a single round. I believe that this paper will become a classic in the literature in the field and I strongly recommend publication. I have a few minor concerns that the authors should consider.

1. Although the authors are rather careful with the language, generally avoiding to be too conclusive, I encourage them to go carefully through the manuscript to avoid making strong statements that may not be sufficiently demonstrated by the data. As an example, in line 365 the authors suggest that '... Type ABC is likely the only functional arrangement ...', referring to which set of NSF subunits may contribute to disassembly through their N-terminal domains. Although I agree with that the data support this conclusion and this sentence is careful enough, the following sentence (line 368) seems to assume that the concept is now well established, which seems premature.

2. The small models depicting the different states generated in the single-molecule force experiments (e.g. fig. 1b,c) do not do a very good job at helping to visualize the states (particularly the half-unzipped state, HU). Moreover, the authors seem to assume that part of the C-terminus of the Q-SNAREs is still structured in the totally unzipped state, TU. However, NMR experiments showed that the C-terminal half of the Q-SNARE motifs become disassembled when the corresponding sequence of synaptobrevin is not present [Trimbuch et al. *elife* 3, e02391 (2014)]. Note also that a fully unstructured C-terminal half of the Q-SNARE complex is more consistent with the small nature of the extension involved in the transition from the TU state to the unstructured coil, UC (Fig. 1c).

3. The small model of Extended Data Fig. 2b is somewhat misleading because this state does not appear to be formed without NSF and α SNAP. Hence, it would be better to include these proteins in the model.

4. The high-speed tracking experiments performed with the magnetic tweezers reveal the intermediate in disassembly mentioned above, which is very interesting. However, it is important to realize that the speed of these experiments is still very slow compared to the rates of some conformational transitions in proteins (e.g. an entire protein may fold in one microsecond). Hence, it would be advisable that the authors warn the readers that there may be additional intermediates in disassembly that the technique cannot detect.

Josep Rizo

Reviewer #2 (Remarks to the Author):

Summary:

The authors use magnetic tweezers to characterize the unfolding landscape of the SNARE complex in the absence and presence of α SNAP and the hexameric motor NSF. They observe that unfolding in the presence of these two factors is faster by an order of magnitude relative to the mechanically

induced unfolding. Moreover, in the presence of α SNAP and NSF the unfolding appears to avoid unfolding intermediates HU and LU observed in the purely mechanical unfolding, as it transitions to the unstructured-coil state (UC). The authors then sought to measure the stoichiometry of α SNAP and NSF assembled on SNARE complexes by using photobleaching analysis. Next, they assembled hybrid NSF motors with fluorescently labeled, N-terminal deleted, mutant protomers doped in. Using photobleaching analysis they measured the maximum number of mutant protomers per hexamer that still support robust 20S complex (SNARE, α SNAP and NSF) formation and disassembly. They also followed the assembly of WT NSF labeled with either donor or acceptor dyes, and by analyzing the FRET histograms they were able to discern the subunit connectivity, and to determine the connectivity of assembled NSF motors with 2 mutant protomers. Moreover, the authors doped in ATP-deficient protomers with Walker B mutations into hybrid NSF hexamers and measured their ATPase rate and compared the hydrolysis rates of free NSF (wild type and doped) and those assembled into the 20S complexes (wild type and doped). Using these observations, they generated a model for the ATP hydrolysis cycle of NSF.

Major concerns:

Most of the characterization of the unfolding signatures in the absence of α SNAP and NSF depicted in figure 1 is straightforward, showing the initial fully zipped (FZ) form, the LU and HU intermediate equilibrium, the totally unfolded (TU) form, and the unstructured coil (UC). However, the interpretation of the intermediate observed in the presence of α SNAP and NSF is less clear.

First, some discussion of the temporal and spatial resolution of the instrument would be helpful given that the representative trace in figure 1e exhibits significant drift while in state FZ. How can the authors be sure that this sort of drift does not account for some of the transients observed in panel h? The unfolding intermediate in panel c (in the absence of α SNAP and NSF) clearly exhibits hopping between HU and LU but is not present in the high-speed transients observed in the presence of α SNAP and NSF. Is it possible that in this latter case this hopping behavior still exists but is sped up dramatically and the observed histogram in panel i represents instead an average between the two states that is unresolvable within the limited temporal resolution of the instrument? While the histogram shows a peak between HU and LU, some of the transients in h appear to dwell in HU. This observation is still consistent with the idea that the intermediates also are present with α SNAP and NSF, but not properly resolved. The authors should address this point

Using fluorescently labeled α SNAP, the authors conclude that α SNAP dimers are the minimal units that connect NSF and SNARE complexes (figure 2). While the data is consistent with this notion it is not conclusive. The high background binding of α SNAP doesn't allow them to rule out that a single α SNAP is sufficient for NSF binding. The authors should provide additional experimental evidence to justify this conclusion.

Critique of Extreme Parsimony Thesis

The authors observe a 20-fold activation in ATPase activity of NSF when complexed in the 20S and suggest that evolutionary pressure evolved NSF to suppress ATPase activity, in order to conserve ATP, when not actively disassembling the SNARE complex. To support this idea, the authors investigate the disassembly activity of hexamers made up of wt and mutant protomers.

The authors assayed hybrid NSF hexamers doped with increasing concentrations of mutant protomers – either with an N-terminal deletion mutant or a Walker B hydrolysis deficient mutant. The authors convincingly show that only a subset of all possible hybrid hexamer compositions can form 20S complexes and are disassembly competent (up to three contiguous mutants). However, the authors do not monitor ATPase hydrolysis of these constructs and therefore it is not clear that this result is evidence of parsimony, namely a selection mechanism that prevents futile ATP hydrolysis from dysfunctional NSF motors. Moreover, the authors do not show that hybrid hexamers disassemble the 20S with the same energetic efficiency as WT.

The authors also find that NSF motors doped with Walker B mutant protomers can assemble 20S complexes but cannot disassemble them, even with a single mutant protomer per hexamer. The authors then measure the bulk ATPase rate of Walker B hybrid hexamers free in solution and in the 20S complex. They interpret the non-linear response in ATPase rate observed with increasing mol fraction of mutant protomers as clear evidence of inter-subunit cooperativity within the NSF while in the 20S complex. Based on these observations, they argue that in the 20S complex, the NSF motor exhibits "...perfect ATP hydrolysis synchronization..." (line 28) and that "...the presence of even a

single [Walker B] subunit prevents ATP hydrolysis of all other five WT subunits" (line 415). However, a non-linear decrease in the activity in the 20 S complex does not imply necessarily that a single mutant monomer in the hexamer is sufficient to abrogate the activity of that hexamer. It is possible to imagine a scenario in which each inactive mutant in the 20S complex alters the activity of its nearest neighbors but not of the other wt monomers, and that more than one such mutant can be incorporated in the ring, and the result will also be a non-linear decrease in the activity of the 20S mixture. Thus, it is my opinion that these results do not support the concept of parsimony either i.e. that complexes that are disassembly incompetent do not hydrolyze ATP above the basal level of hydrolysis. Moreover, a Walker B mutation generally abolishes nucleotide hydrolysis while still retaining a tightly bound ATP. Therefore, it is possible that a hybrid hexamer containing Walker B mutations cannot disassemble the SNARE complex not because the WT monomer cannot hydrolyze ATP but because the mutant protomers cannot transition to a disassembly competent state downstream of ATP binding. Moreover, the authors claim that SNARE disassembly occurs with "minimal dissipation" (line 530), but they have not measured the energy efficiency of the complex and any such statement is not justified.

The authors clearly show stimulated ATP hydrolysis with positive cooperativity of NSF in the 20S complex when compared to the negative cooperativity and slow ATP hydrolysis of free NSF. Based on these observations they build a mechanochemical model for the ATPase ring in the 20S complex. However, no attempt is made to determine where in the cycle binding and hydrolysis of ATP, as well as ADP and phosphate release occur relative to the disassembly step. To formulate such mechanochemical cycle the authors should provide direct experimental evidence of the occurrence of various chemical and mechanical steps.

Assessment:

Although the authors have convincingly demonstrated cooperativity between the subunits and α SNAP and SNARE stimulated ATP hydrolysis, the central thesis as written, that of extreme parsimony with regards to ATP consumption, is not well supported and should be either reassessed or validated by experimental data, before the paper can be accepted for publication.

Minor concerns:

1. In the magnetic tweezers experiment VAMP2 and Syntaxin-1A appear to be a linear fusion as indicated by the cartoon of the UC state in panel c, otherwise the tether would break. The nature of this construct was not clear from the text or methods section – as written it seems like two separate proteins.
2. The authors performed the magnetic tweezers experiments at saturating ATP concentrations. Based on bulk ATPase measurements, the ATP hydrolysis rate of NSF in the 20S complex is around 1 ATP/sec, thus SNARE disassembly would require at least 6 sec (or 6 ATPs). However, at 16pN, the FZ state dwell time is 2 sec. Does the later mean that at 16pN NSF can disassemble SNARE while hydrolyzing only 2 ATPs?
3. How are the fit curves in figure 2e and 5f generated? Please, provide the functional form for clarity.
4. Regarding line 132-135, "...suggesting substantial energy is required to initiate disassembly" Do the authors mean activation energy? Please, clarify.
5. In line 527, the sentence "Our observation... suggests a strong mechanical coupling ... that delivers mechanical tension to all four SNARE motifs with minimal dissipation" it is not clear that NSF must deliver mechanical tension to all four SNARE motifs. For example, in the magnetic tweezers assay, in the absence of α SNAP and NSF, the SNARE complex is unfolded by pulling on just two of the four SNARE motifs.
6. Line 298-299. "we still found that these hybrid hexamers directed SNARE complex disassembly as efficiently as WT hexamers". The word "efficiently" is ambiguous here and could perhaps be changed to read "to the same extent...".
7. Line 583-585: "Once fully loaded with ATP molecules... NSF synchronizes all six subunits to induce hydrolysis of the bound ATPs at a rate one order of magnitude higher than the random ATP hydrolysis rate" Therefore, the two modes of ATP hydrolysis, slow (γ) and fast (β), are mutually exclusive. However, the authors consider both modes in their model (Figure 6a). Why? Please, clarify.
8. For ATP hydrolysis, free NSF exhibits a Hill coefficient of $n = 0.58$. Therefore, the authors consider a scaling factor α to account for the negative cooperativity of ATP binding in their model (Figure 6d). However, the authors do not consider a scaling factor in the presence of α SNAP and SNARE, although it exhibits a Hill coefficient of $n = 1.3$. What is the rationale for deciding whether to use α or

not?

9. The ATPase assay is performed using excess of α SNAP over NSF and SNARE, implying that two complexes are formed in the reaction: 20S and α SNAP-NSF. What is the ATP hydrolysis rate of α SNAP-NSF and how does this affect the model?

Clerical issues and typos:

1. In the text, the intermediate state observed during the 20S complex-mediated disassembly is referred to as I20S, but in Figure 1, it is labelled as INT. Please, correct or clarify.

2. In line 391, the authors use the phrase “with ATP hydrolysis permitted...” which was confusing.

Perhaps it would be more straightforward to write something along the lines of “in the presence/absence of EDTA, a chelator of magnesium that inhibits hydrolysis...” or something to that effect.

3. In line 510, the authors use the phrase “concentrate their energy expenditure...” which is confusing and should be rewritten.

4. Line 483-485, β refers to stimulated hydrolysis rate per promoter. However, in Figure 6a, β seems to refer to the ATP hydrolysis rate of the full NSF ring. Please, clarify.

Point-by-point response to the reviewer comments on “Extreme parsimony in ATP consumption by 20S complexes in the global disassembly of single SNARE complexes”

Reviewer #1

This beautiful paper describes an analysis of the mechanism of disassembly of the neuronal SNARE complex by NSF and α SNAP using single-molecule force measurements in combination with single molecule fluorescence spectroscopy. Among multiple interesting results, the authors reveal an intermediate state in the disassembly pathway that reflects the initial step that leads to disassembly and likely involves opening of the C-terminus of the SNARE four-helix due to interactions with α SNAP. They also show that a minimum of two α SNAP molecules and three NSF N-terminal domains are required for disassembly, that such minimal 20S complexes are almost as efficient as WT 20S complex, and that the random nature of ATP binding to individual NSF subunits is converted to an almost perfect synchronization in ATP hydrolysis that results in the disassembly of the SNARE complex in a single round. I believe that this paper will become a classic in the literature in the field and I strongly recommend publication. I have a few minor concerns that the authors should consider.

Response) We thank the reviewer for the positive reception of our work.

1. Although the authors are rather careful with the language, generally avoiding to be too conclusive, I encourage them to go carefully through the manuscript to avoid making strong statements that may not be sufficiently demonstrated by the data. As an example, in line 365 the authors suggest that ‘... Type ABC is likely the only functional arrangement ...’, referring to which set of NSF subunits may contribute to disassembly through their N-terminal domains. Although I agree with that the data support this conclusion and this sentence is careful enough, the following sentence (line 368) seems to assume that the concept is now well established, which seems premature.

Response) We thank the reviewer for the careful consideration and revised the manuscript in several places. For example:

[Main Text p14] “Our approach with single-protomer resolution suggests that three neighboring NSF subunits and an α SNAP dimer define the minimal unit for 20S complex formation.”

[Main Text p20] “Our single-molecule magnetic tweezer data suggested that the 20S complex induces efficient and simultaneous disassembly of all four SNARE motifs in the SNARE complex on millisecond time scales.”

2. The small models depicting the different states generated in the single-molecule force experiments (e.g. fig. 1b,c) do not do a very good job at helping to visualize the states (particularly the half-unzipped state, HU). Moreover, the authors seem to assume that part of the C-terminus of the Q-SNAREs is still structured in the totally unzipped state, TU. However, NMR experiments showed that the C-terminal half of the Q-SNARE motifs become disassembled when the corresponding sequence of synaptobrevin is not present [Trimbuch et al. *elife* 3, e02391 (2014)]. Note also that a fully unstructured C-terminal half of the Q-SNARE complex is more consistent with the small nature of the extension involved in the transition from the TU state to the unstructured coil, UC (Fig. 1c).

Response) We agree with the reviewer that the cartoons depicting the conformational states of SNARE complexes had many rooms for improvement. To improve the models in terms of clarity, we simplified the cartoons in Fig. 1b,c and Supplementary Fig. 2b by relaxing the twisted structures of the bundles. Additionally, the models for HU and TU were updated to reflect the reviewer’s comment on the unstructured nature of the C-terminal regions. We note that the new calculations (C-terminal SNARE motifs unstructured up to +4 and +2 layers for HU and TU, respectively) slightly increase the extension levels of HU and TU in Fig. 1 and Supplementary Figs. 1 and 2, but the main conclusions essentially remain valid. In fact, with the new HU model, the intermediate state (I_{20s}) in the presence of NSF became even less likely to correspond to HU observed in purely mechanical unfolding. We once again thank the reviewer for the valuable comments. We revised the manuscript as below, and added the mentioned reference to support the revision:

[Main text, p7] “The extension level of TU was well explained by the Q-SNARE model with unstructured C-terminal motifs (up to +2 layer) in agreement with the results from NMR experiments^{6,8,35}.”

35. Trimbuch, T. et al. Re-examining how complexin inhibits neurotransmitter

release. *eLife* **3**, e02391 (2014).

3. The small model of Extended Data Fig. 2b is somewhat misleading because this state does not appear to be formed without NSF and α SNAP. Hence, it would be better to include these proteins in the model.

Response) We added the cartoons of NSF and α SNAP in Supplementary Fig. 2b.

4. The high-speed tracking experiments performed with the magnetic tweezers reveal the intermediate in disassembly mentioned above, which is very interesting. However, it is important to realize that the speed of these experiments is still very slow compared to the rates of some conformational transitions in proteins (e.g. an entire protein may fold in one microsecond). Hence, it would be advisable that the authors warn the readers that there may be additional intermediates in disassembly that the technique cannot detect.

Response) We agree with the reviewer, and added the following sentences in the revision:

[Main text, p8] “We noticed that any short-lived intermediates below hundreds of milliseconds would not be resolved in our measurement with a time resolution of 0.83 ms. We cannot exclude a possibility that the I_{20s} state might reflect an averaged-out level as a result of rapid transitions between LU and HU. Nevertheless, the symmetric unfolding model remains valid, in which a resulting HU state is estimated to have Syx unfolded up to the +3 layer. Thus, the disparity between I_{20s} and HU turns out to be small in the symmetric unfolding, measuring only an one-layer difference.”

Point-by-point response to the reviewer comments on “Extreme parsimony in ATP consumption by 20S complexes in the global disassembly of single SNARE complexes”

Reviewer #2

Summary:

The authors use magnetic tweezers to characterize the unfolding landscape of the SNARE complex in the absence and presence of α SNAP and the hexameric motor NSF. They observe that unfolding in the presence of these two factors is faster by an order of magnitude relative to the mechanically induced unfolding. Moreover, in the presence of α SNAP and NSF the unfolding appears to avoid unfolding intermediates HU and LU observed in the purely mechanical unfolding, as it transitions to the unstructured-coil state (UC). The authors then sought to measure the stoichiometry of α SNAP and NSF assembled on SNARE complexes by using photobleaching analysis. Next, they assembled hybrid NSF motors with fluorescently labeled, N-terminal deleted, mutant protomers doped in. Using photobleaching analysis they measured the maximum number of mutant protomers per hexamer that still support robust 20S complex (SNARE, α SNAP and NSF) formation and disassembly. They also followed the assembly of WT NSF labeled with either donor or acceptor dyes, and by analyzing the FRET histograms they were able to discern the subunit connectivity, and to determine the connectivity of assembled NSF motors with 2 mutant protomers. Moreover, the authors doped in ATP-deficient protomers with Walker B mutations into hybrid NSF hexamers and measured their ATPase rate and compared the hydrolysis rates of free NSF (wild type and doped) and those assembled into the 20S complexes (wild type and doped). Using these observations, they generated a model for the ATP hydrolysis cycle of NSF.

Response) We thank the reviewer for summarizing the main observations of our work.

Major concerns:

Most of the characterization of the unfolding signatures in the absence of α SNAP and NSF depicted in figure 1 is straightforward, showing the initial fully zipped (FZ) form, the LU and

HU intermediate equilibrium, the totally unfolded (TU) form, and the unstructured coil (UC). However, the interpretation of the intermediate observed in the presence of α SNAP and NSF is less clear.

First, some discussion of the temporal and spatial resolution of the instrument would be helpful given that the representative trace in figure 1e exhibits significant drift while in state FZ. How can the authors be sure that this sort of drift does not account for some of the transients observed in panel h?

Response) We would like to draw the reviewer's attention to Supplementary Fig. 1a, in which we analyzed the spatiotemporal resolution of our magnetic tweezer apparatus in terms of the Allan deviation. The analysis shows that a resolution of ~ 1 nm is achieved if the original 1.2-kHz trace is averaged with an 100 ms window. Thus, the fluctuations on a few second scales observed in Fig. 1e are not apparatus-induced drifts, but likely represent molecular dynamics of the tweezed SNARE proteins.

We also note that on the 10 ms scale, close to lifetimes of the 20S complex-specific intermediate (I_{20s}), the Allan deviation is increased to around 2 to 3 nm, yet smaller than ~ 20 nm observed for the extension changes with the I_{20s} transitions (Fig. 1h,i). Thus, it is unlikely that the baseline fluctuations (hopping between FZ and LU) are responsible for the observed intermediates during NSF-mediated disassembly.

The unfolding intermediate in panel c (in the absence of α SNAP and NSF) clearly exhibits hopping between HU and LU but is not present in the high-speed transients observed in the presence of α SNAP and NSF. Is it possible that in this latter case this hopping behavior still exists but is sped up dramatically and the observed histogram in panel **I** represents instead an average between the two states that is unresolvable within the limited temporal resolution of the instrument? While the histogram shows a peak between HU and LU, some of the transients in h appear to dwell in HU. This observation is still consistent with the idea that the intermediates also are present with α SNAP and NSF, but not properly resolved. The authors should address this point

Response) We agree with the reviewer, and added the following sentences in the revision:

[Main text, p8] "We noticed that any short-lived intermediates below hundreds of milliseconds would not be resolved in our measurement with a time resolution of 0.83 ms. We cannot exclude a possibility that the I_{20s} state might reflect an averaged-

out level as a result of rapid transitions between LU and HU. Nevertheless, the symmetric unfolding model remains valid, in which a resulting HU state is estimated to have Syx unfolded up to the +3 layer. Thus, the disparity between I_{20s} and HU turns out to be small in the symmetric unfolding, measuring only an one-layer difference.”

Using fluorescently labeled α SNAP, the authors conclude that α SNAP dimers are the minimal units that connect NSF and SNARE complexes (figure 2). While the data is consistent with this notion it is not conclusive. The high background binding of α SNAP doesn't allow them to rule out that a single α SNAP is sufficient for NSF binding. The authors should provide additional experimental evidence to justify this conclusion.

Response) We understand the reviewer's concern that a single α SNAP might be sufficient for NSF binding. To address this issue, we carried out additional experiments where we fluorescently labeled both α SNAPs and NSFs with different dyes of Alexa 647 and DY549, respectively. We narrowed down our photobleaching analysis to the spots that showed both Alexa 647 and DY549 fluorescence signals, thus to those with legitimate 20S complex assembly.

Remarkably, with this new photobleaching analysis restricted to co-localized spots, populations showing a single photobleaching step almost completely disappeared (Fig. 2c,d, pasted below). An absolute majority of the co-localized spots showed at least two photobleaching steps. We also found that a meaningful fraction of 20S complexes contained three copies of α SNAPs, confirming our conclusion that the minimal structural unit for 20S complex assembly is α SNAP dimers.

Figure 2c,d

Critique of Extreme Parsimony Thesis.

The authors observe a 20-fold activation in ATPase activity of NSF when complexed in the 20S and suggest that evolutionary pressure evolved NSF to suppress ATPase activity, in order to conserve ATP, when not actively disassembling the SNARE complex. To support this idea, the authors investigate the disassembly activity of hexamers made up of wt and mutant protomers. The authors assayed hybrid NSF hexamers doped with increasing concentrations of mutant protomers – either with an N-terminal deletion mutant or a Walker B hydrolysis deficient mutant. The authors convincingly show that only a subset of all possible hybrid hexamer compositions can form 20S complexes and are disassembly competent (up to three contiguous mutants). However, the authors do not monitor ATPase hydrolysis of these constructs and therefore it is not clear that this result is evidence of parsimony, namely a selection mechanism that prevents futile ATP hydrolysis from dysfunctional NSF motors. Moreover, the authors do not show that hybrid hexamers disassemble the 20S with the same energetic efficiency as WT.

Response) We agree with the reviewer that a more stringent test of our hypothesis will be a direct measurement of ATP consumption by the mutant NSF hexamers. As we showed in Figure 3, the N-MT hybrid hexamers showed an much mitigated avidity for 20S complex formation compared with the WT hexamers. It was thus difficult to selectively enrich 20S complexes with three N-MT subunits, a technical challenge for measuring the ATP consumption rates specific to the N-MT hybrid hexamers.

We instead took advantage of the single-molecule resolution of our assay in monitoring SNARE complex disassembly. With a mixture of hybrid N-MT hexamers, in which ~20% of NSF hexamers contained three N-MT subunits (Fig. 3c), we previously demonstrated that the resulting 20S complexes fully retained the capability of SNARE complex disassembly within 5 minutes. By taking one step further, we determined detailed kinetics of this disassembly reaction (Fig. 3f). The new results, added as Fig. 3f in the revised manuscript, clearly show that the N-MT hybrid NSF hexamers disassembled the single SNARE complexes as rapidly as purely WT NSF hexamers. In addition, we needed only a single, not two, exponential function to the disassembly reaction for the hybrid N-MT

hexamers with a time constant of 30.7 ± 3.3 s, which fairly fell within a same range as 28.4 ± 2.9 s determined for the WT hexamers. These data corroborate our claim that all the 20S complexes assembled with N-MT hybrid hexamers are as fully functional as those with WT NSF hexamers in terms of the SNARE complex disassembly. Equivalently, these data support our idea that any configurations that do not support a disassembly activity have been excluded from 20S complex assembly, probably through evolution, which hints about the resulting parsimony in ATP consumption by the 20S complexes.

The authors also find that NSF motors doped with Walker B mutant protomers can assemble 20S complexes but cannot disassemble them, even with a single mutant protomer per hexamer. The authors then measure the bulk ATPase rate of Walker B hybrid hexamers free in solution and in the 20S complex. They interpret the non-linear response in ATPase rate observed with increasing mol fraction of mutant protomers as clear evidence of inter-subunit cooperativity within the NSF while in the 20S complex. Based on these observations, they argue that in the 20S complex, the NSF motor exhibits “...perfect ATP hydrolysis synchronization...” (line 28) and that “...the presence of even a single [Walker B] subunit prevents ATP hydrolysis of all other five WT subunits” (line 415). However, a non-linear decrease in the activity in the 20S complex does not imply necessarily that a single mutant monomer in the hexamer is sufficient to abrogate the activity of that hexamer. It is possible to imagine a scenario in which each inactive mutant in the 20S complex alters the activity of its nearest neighbors but not of the other wt monomers, and that more than one such mutant can be incorporated in the ring, and the result will also be a non-linear decrease in the activity of the 20S mixture. Thus, it is my opinion that these results do not support the concept of parsimony either i.e. that complexes that are disassembly incompetent do not hydrolyze ATP above the basal level of hydrolysis.

Response) The reviewer raised an alternative hypothesis that each mutant subunit alters the activity of only the nearest neighbors. We thank the reviewer for this comment, which gives us to examine our hypothesis in a novel angle that has not been examined before.

We first considered our observations in Fig. 5b,c that incorporation of even a single A-MT subunit in the NSF hexamer totally

Figure 5b,c

abolished the disassembly activity, which is likely at odds with the hypothesis that a mutant subunit only affect neighboring ones.

In addition, we carried out a further analysis for the non-linear decrease in ATP hydrolysis rate observed for the 20S complexes as more A-MT subunits were included (Fig. 5e,f). We considered different scenarios in which different numbers of A-MT subunits are required to eliminate all ATP hydrolysis activity of the 20S complexes. The hypothesis mentioned in the reviewer's comment would correspond to a weaker version of a scenario that inclusion of two A-MTs quenches the ATP hydrolysis activity of entire NSF hexamers (because if A-MT subunits are positioned adjacent to one another, more than two A-MTs would be required to see complete inhibition of ATP hydrolysis). For clarity, we assumed that two A-MT subunits abolish the ATP hydrolysis activity regardless of their relative positions.

Figure 5e,f

Remarkably, we found that even this stringent scenario assuming two A-MT subunits failed to explain the observed non-linear decrease for the 20S complex with A-MT hybrid hexamers. These data strongly suggest that inclusion of a single A-MT subunit eliminate the stimulated ATPase activity of the entire NSF hexamer. Altogether, in the stimulated ATPase activity of the NSF hexamer proven to be indispensable for the disassembly activity, the six subunits are strongly coordinated to fire at the same time.

Moreover, a Walker B mutation generally abolishes nucleotide hydrolysis while still retaining a tightly bound ATP. Therefore, it is possible that a hybrid hexamer containing Walker B mutations cannot disassemble the SNARE complex not because the WT monomer cannot hydrolyze ATP but because the mutant protomers cannot transition to a disassembly

competent state downstream of ATP binding.

Response) This is the question we asked ourselves when we observed the data in Fig. 5b,c, which collectively suggested that inclusion of a single A-MT subunit abolishes the disassembly activity of the whole 20S complexes. An immediate question is whether the ATPase activity itself is inhibited or any conformational changes, downstream to ATP hydrolysis, are inhibited. We examined the ATPase activity of the A-MT hybrid NSF hexamers, and observed the steep non-linear decrease as a function of molar fractions of A-MT subunits. This observation clearly indicates that it is the ATP hydrolysis step that is directly inhibited in the presence of A-MT subunits.

We further found that this observation of non-linear decrease is selective to the stimulated ATP-hydrolysis mode. The basal mode of ATP hydrolysis occurs in six NSF subunits in a random manner, thus persisting in the WT subunits of the A-MT hybrid hexamers loaded to form the 20S complexes.

Moreover, the authors claim that SNARE disassembly occurs with “minimal dissipation” (line 530), but they have not measured the energy efficiency of the complex and any such statement is not justified.

Response) Following the reviewer comment, we removed the words in the revised manuscript.

The authors clearly show stimulated ATP hydrolysis with positive cooperativity of NSF in the 20S complex when compared to the negative cooperativity and slow ATP hydrolysis of free NSF. Based on these observations they build a mechanochemical model for the ATPase ring in the 20S complex. However, no attempt is made to determine where in the cycle binding and hydrolysis of ATP, as well as ADP and phosphate release occur relative to the disassembly step. To formulate such mechanochemical cycle the authors should provide direct experimental evidence of the occurrence of various chemical and mechanical steps.

Response) We agree with the reviewer comment that despite many experimental data, the current manuscript still lacks direct evidence that the mechanical disassembly activity coincides with stimulated ATP hydrolysis. In the revised manuscript, we removed the terms of “mechanochemical cycles” and instead used a more neutral term of “ATP hydrolysis

cycles”. In the main text, we stated the possibility that the stimulated ATP hydrolysis would align with the disassembly activity.

It is still an open question which step in the stimulated ATP hydrolysis exactly defines the disassembly moment. Our previous experiments using phosphate ion analogs (*Science* (2015) 347, 1485-9) indicated that the disassembly activity should tail release of phosphate ions. This would potentially place ADP-bound state or ADP release for the candidate state defining the disassembly step. Dissection of this mechanochemical cycle in a great detail is a remaining important question in the study of 20S complexes and warrants further investigation.

Assessment:

Although the authors have convincingly demonstrated cooperativity between the subunits and α SNAP and SNARE stimulated ATP hydrolysis, the central thesis as written, that of extreme parsimony with regards to ATP consumption, is not well supported and should be either reassessed or validated by experimental data, before the paper can be accepted for publication.

Response) We thank the reviewer for critical assessment of our data and claims. With the additional experimental data and analysis, we hope the reviewer finds that our conclusions in the revised manuscript are much more strongly supported by the experimental data.

Minor concerns:

1. In the magnetic tweezers experiment VAMP2 and Syntaxin-1A appear to be a linear fusion as indicated by the cartoon of the UC state in panel c, otherwise the tether would break. The nature of this construct was not clear from the text or methods section – as written it seems like two separate proteins.

Response) We modified the figure panel to highlight that VAMP2 and Syntaxin-1A are knotted via a disulfide bond introduced near their N-terminal ends. We also edited the Method section accordingly.

[Method] “For magnetic tweezer experiment, N-terminal His-tagged cysteine free SNAP-25A, synaptobrevin-2 (2-97, L32C/I97C) and syntaxin-1A (191-268, I202C/K266C) were cloned into pET28a vector. Among four cysteines, two

cysteines (I97C in synaptobrevin-2, K277C in syntaxin-1A) were used to conjugate the DNA handle for magnetic tweezer experiments. The other two cysteines were used to form a disulfide bond for repeated magnetic tweezer experiment”

2. The authors performed the magnetic tweezers experiments at saturating ATP concentrations. Based on bulk ATPase measurements, the ATP hydrolysis rate of NSF in the 20S complex is around 1 ATP/sec, thus SNARE disassembly would require at least 6 sec (or 6 ATPs). However, at 16pN, the FZ state dwell time is 2 sec. Does the later mean that at 16pN NSF can disassemble SNARE while hydrolyzing only 2 ATPs?

Response) In Fig. 6, the ATP hydrolysis rates are normalized values per protomer per min. Thus, the actual rate of ATP hydrolysis by a single ‘NSF hexamer’ in the 20S complex will be ~6 ATP/sec. In this sense, the dwell time in the FZ state before the disassembly event, on the order of a few seconds (Fig. 1f), reasonably matches with the latency time for stimulated ATP hydrolysis estimated from Figs. 5 and 6.

3. How are the fit curves in figure 2e and 5f generated? Please, provide the functional form for clarity.

Response) We removed Fig. 2e in the revised manuscript because we added a new set of photobleaching analysis performed under co-localization conditions. The curves in figure 5f are theoretical lines following equation edited in the Method section.

[Method] “The theoretical lines in Fig. 5f are following the equations below.

$$y = (1 - x) \sum_{i=0}^{j-1} \binom{5}{i} (1 - x)^{5-i} x^i$$

where x is mutant fraction and j is the number of mutant subunits per hexamer required to eliminate function.”

4. Regarding line 132-135, “...suggesting substantial energy is required to initiate disassembly” Do the authors mean *activation energy*? Please, clarify.

Response) It can be viewed as the activation energy for opening the linker regions of the SNARE motifs, but we prefer not to designate it so because there are multiple energy barriers

throughout the reaction coordinate. We revised the manuscript as follows:

[Main text, p5] “Up to forces of 12 pN, the SNARE complexes including their linker regions remained “fully zippered” (the FZ state), suggesting a substantial energy barrier to initiating disassembly.”

5. In line 527, the sentence “*Our observation... suggests a strong mechanical coupling ... that delivers mechanical tension to all four SNARE motifs with minimal dissipation*” it is not clear that NSF must deliver mechanical tension to all four SNARE motifs. For example, in the magnetic tweezers assay, in the absence of α SNAP and NSF, the SNARE complex is unfolded by pulling on just two of the four SNARE motifs.

Response) As the reviewer pointed out, the mechanical disassembly by magnetic tweezers was enacted by pulling on two strands. As a result, we observed the “residual SNAP-25” signature shown in Fig. 1c, a long latency time before SNAP-25 dissociation (transition from TU to UC). However, the NSF-mediated disassembly exhibited direct transition to UC, without any detectable dwelling in TU. This observation likely indicates that in addition to VAMP2 and syntaxin-1A proteins on to which DNA handles are directly attached, the rest two of the four strands (i.e. two SNARE motifs of SNAP-25) were also being unraveled at the same time in the 20S complex-mediated disassembly within our time resolution (~a few ms).

6. Line 298-299. “we still found that these hybrid hexamers directed SNARE complex disassembly as efficiently as WT hexamers”. The word “efficiently” is ambiguous here and could perhaps be changed to read “to the same extent...”.

Response) We edited the words following the reviewer comment.

7. Line 583-585: “*Once fully loaded with ATP molecules... NSF synchronizes all six subunits to induce hydrolysis of the bound ATPs at a rate one order of magnitude higher than the random ATP hydrolysis rate*” Therefore, the two modes of ATP hydrolysis, slow (γ) and fast (β), are mutually exclusive. However, the authors consider both modes in their model (Figure 6a). Why? Please, clarify.

Response) We found the basal ATPase activity, albeit slow (represented by a rate of γ),

persisted for the NSF hexamers loaded in the 20S complexes (Fig. 5g-i). This basal mode of ATP hydrolysis, occurring in each subunit independent of others, randomly consumes one ATP molecule in one of the subunits, thereby decreasing the number of bound ATP molecules by one (i.e., from n to $n-1$). When a NSF hexamer happens to be loaded with full six ATP molecules, the synchronized mode of ATP hydrolysis becomes allowed, which consumes all bound ATP molecules at an increased rate of β . Therefore, rather than being mutually exclusive to one another, the two modes of ATP hydrolysis work in concert, with the basal hydrolysis mode primarily directing ATP binding and the synchronized hydrolysis mode driving the mechanical disassembly of the rigid substrate of SNARE complex. As discussed in our Discussion section, adoption of such random ATP binding might have been an ideal adaptation for NSF that has much longer idle times between synchronized ATP hydrolysis than ClpXP and $\phi 29$ motors that should go through many consecutive cycles of synchronized ATP hydrolysis without intermittent pause times.

8. For ATP hydrolysis, free NSF exhibits a Hill coefficient of $n = 0.58$. Therefore, the authors consider a scaling factor α to account for the negative cooperativity of ATP binding in their model (Figure 6d). However, the authors do not consider a scaling factor in the presence of α SNAP and SNARE, although it exhibits a Hill coefficient of $n = 1.3$. What is the rationale for deciding whether to use α or not?

Response) This is a great suggestion. Following the reviewer comment, we tried fit the ATP titration curve of the 20S complexes with KNF model containing a scaling factor α .

Remarkably, the fitting yielded a scaling factor very close to 1 ($\alpha=1.1\pm 0.1$), reducing the

KNF model to the single K_d model originally used. This result supports the validity of the MWC model we used to explain the ATP hydrolysis cycle of the NSF hexamer loaded in 20S complexes. This new analysis results were included as new panels in Supplementary Figure 8c,e and we revised the manuscript as below.

[Main text, p19] “We applied the KNF model to the ATP titration curve of the 20S complexes in Fig. 6b,c, and found that the theoretical model converged once again to the simpler model in Fig. 6a with a scaling factor of nearly 1, assuring the validity of the ATP hydrolysis cycle we constructed for the 20S complexes (Supplementary Fig. 8c,e).”

9. The ATPase assay is performed using excess of α SNAP over NSF and SNARE, implying that two complexes are formed in the reaction: 20S and α SNAP-NSF. What is the ATP hydrolysis rate of α SNAP-NSF and how does this affect the model?

Response) We measured the ATPase activity of NSF with only α SNAP. We added the result in the supplementary figure (Supplementary Fig. 8i). The result shows that the ATPase activity of NSF was increased by α SNAP addition. Moreover, the increase was similar to the difference in γ in figure 6.

Clerical issues and typos:

1. In the text, the intermediate state observed during the 20S complex-mediated disassembly is referred to as I_{20S} , but in Figure 1, it is labelled as INT. Please, correct or clarify.

Response) We apologize for this mistake. Throughout the revised manuscript, we used the term of “ I_{20S} ” to denote the intermediate observed during the 20S complex-mediated SNARE complex disassembly.

2. In line 391, the authors use the phrase “*with ATP hydrolysis permitted...*” which was confusing. Perhaps it would be more straightforward to write something along the lines of “in the presence/absence of EDTA, a chelator of magnesium that inhibits hydrolysis...” or something to that effect.

Response) We have edited the manuscript accordingly. We now use the words “under ATP-hydrolyzing condition” and “under non-ATP hydrolyzing condition” to denote the conditions in a more explicit manner.

3. In line 510, the authors use the phrase “concentrate their energy expenditure...” which is

confusing and should be rewritten.

Response) In the revised manuscript, the phrase was revised to “intensively consume energy in to the functional 20S complex”.

4. Line 483-485, β refers to stimulated hydrolysis rate per promoter. However, in Figure 6a, β seems to refer to the ATP hydrolysis rate of the full NSF ring. Please, clarify.

Response) We apologize for confusion. For direct comparison, the two rates γ and β have been normalized per protomer and unit time. However, the coupled ATP hydrolysis rate is also constant for hexamer because all ATPs are induced simultaneously. We revised the legend of Figure 6 and described it in Method section.

Reviewer #1 (Remarks to the Author):

The authors have addressed my concerns and I recommend acceptance without further changes.

Reviewer #2 (Remarks to the Author):

I have reviewed the answers of the authors to each of the questions I raised and I believe that they have responded satisfactorily each of those questions. Therefore, I believe that the paper should now be accepted for publication in Nature Communications.